# Smoothed Preference Optimization via ReNoise Inversion for Aligning Diffusion Models with Varied Human Preferences

Yunhong Lu [1]   Qichao Wang [1]   Hengyuan Cao [1]   Xiaoyin Xu [2]   Min Zhang [1 3]

## Abstract

Direct Preference Optimization (DPO) aligns text-to-image (T2I) generation models with human preferences using pairwise preference data. Although substantial resources are expended in collecting and labeling datasets, a critical aspect is often neglected: *preferences vary across individuals and should be represented with more granularity.* To address this, we propose SmPO-Diffusion, a novel method for modeling preference distributions to improve the DPO objective, along with a numerical upper bound estimation for the diffusion optimization objective. First, we introduce a smoothed preference distribution to replace the original binary distribution. We employ a reward model to simulate human preferences and apply preference likelihood averaging to improve the DPO loss, such that the loss function approaches zero when preferences are similar. Furthermore, we utilize an inversion technique to simulate the trajectory preference distribution of the diffusion model, enabling more accurate alignment with the optimization objective. Our approach effectively mitigates issues of excessive optimization and objective misalignment present in existing methods through straightforward modifications. Experimental results demonstrate that our method achieves state-of-the-art performance in preference evaluation tasks, surpassing baselines across various metrics, while reducing the training costs.

## 1 Introduction

T2I diffusion models (Rombach et al., 2021; Podell et al., 2023) have recently gained widespread popularity. However,

several challenges remain, such as limited text rendering capabilities (Chen et al., 2023a), unrealistic spatial layouts (Lin et al., 2024) and improper illumination (Ren et al., 2024). Current approaches to addressing these challenges primarily focus on optimizing training strategies (Esser et al., 2024), improving network architectures (Peebles & Xie, 2022), augmenting datasets (Gadre et al., 2023) and incorporating richer semantic conditions (Chen et al., 2024; Pernias et al., 2023). Nevertheless, these improvements typically necessitate retraining the models from scratch, making them unsuitable for enhancing pre-existing models. Inspired by reinforcement learning with human feedback (RLHF) methods used in large language models (LLMs) (Wang et al., 2024), aligning T2I models with human preferences has emerged as a promising direction to improve model performance (Liu et al., 2024), though it is currently under active exploration and holds significant potential.

Current research primarily focuses on two directions. The *first* involves collecting human-labeled preference images to train models (Lee et al., 2023b; Liang et al., 2023; Dai et al., 2023). These datasets typically assume a binary preference distribution (Kirstain et al., 2023), which not only requires substantial human resources but also neglects the inherent variability of human preferences, often resulting in **excessive optimization**. The *second* direction focuses on designing methods to fine-tune T2I models using human feedback data. Many previous approaches propagate reward maximization through reward models (Clark et al., 2023; Prabhudesai et al., 2023), leading to reward hacking. Alternatively, some approaches model the sampling process of diffusion models as a Markov chain and employ reinforcement learning (RL) objectives based on reward feedback (Black et al., 2023; Fan et al., 2023; Zhang et al., 2024b; Li et al., 2024b). However, these methods require extensive online evaluations and suffer from training instability. Moreover, DPO methods (Wallace et al., 2023; Yang et al., 2023; 2024) optimize the RL objective toward the optimal trajectory strategy for diffusion models. Nevertheless, estimation based on the forward process often results in **misalignment** between the optimization objective and the desired outcome.

To address these issues, we propose an *adaptive* (with image-dependent loss), and *efficient* approach for preference alignment of T2I diffusion models. **Our insight is that, within**

---

[1]College of Computer Science and Technology, Zhejiang University. [2]College of Biomedical Engineering and Instrument Science, Zhejiang University. [3]Shanghai Institute for Advanced Study-Zhejiang University. Correspondence to: Min Zhang <min_zhang@zju.edu.cn>.

*Proceedings of the 42$^{nd}$ International Conference on Machine Learning*, Vancouver, Canada. PMLR 267, 2025. Copyright 2025 by the author(s).

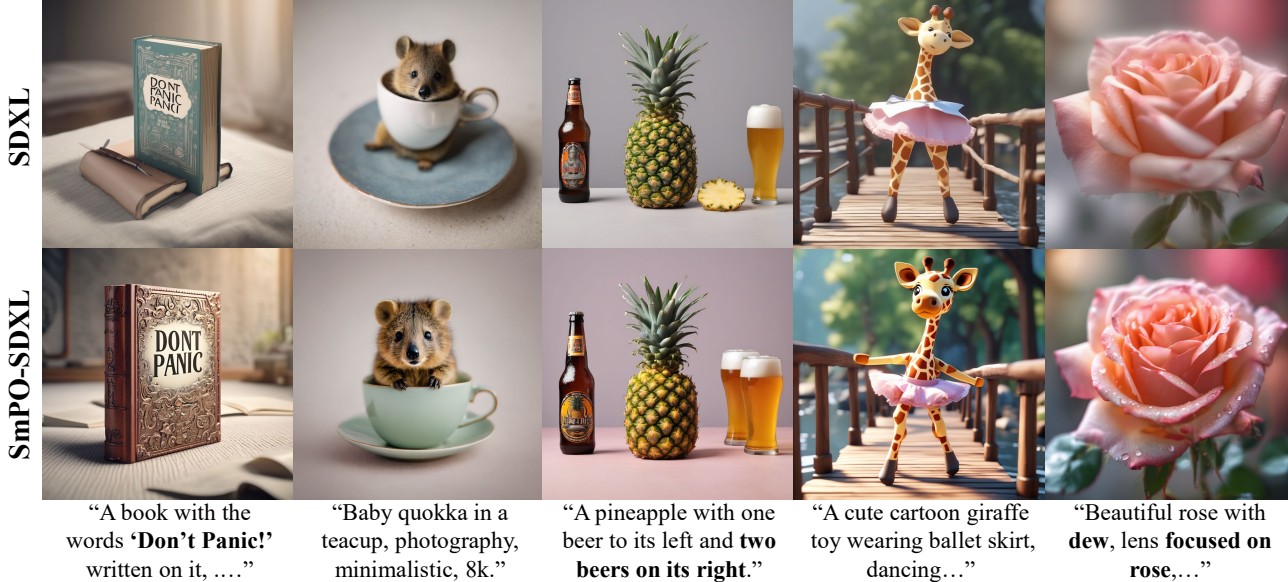

Figure 1: **Enhancement over Vanilla SDXL**. We introduce SmPO-Diffusion, designed to address the variability of human preferences, as an adaptive and efficient approach to aligning diffusion models with human preferences. The images in the first row are generated by the SDXL model, while the second row presents images generated by the SDXL model fine-tuned using SmPO-Diffusion (denoted as SmPO-SDXL). The results demonstrate that SmPO-SDXL generates higher-quality images, with significant enhancements in spatial layouts, text rendering, proper illumination and visual appeal.

**the diffusion model framework, existing methods exhibit significant inaccuracies in both modeling human preferences and their estimation techniques.** Thus, we introduce SmPO-Diffusion for modeling preference distributions to improve the DPO loss function, along with a numerical upper bound estimation for the diffusion optimization objective. To this end, we introduce two key contributions:

**Smoothed Preference Modeling.** As the adage goes, "Beauty is in the eye of the beholder." Given the inherent variability in human preferences, we propose the use of smoothed preference distributions to replace binary preferences. This approach mitigates label bias by integrating the average likelihood of preferences into the DPO loss function, as the loss function approaches zero when preferences are similar. In practice, our method eliminates the need for manual annotation by automatically generating smooth preferences through a reward model, significantly reducing data collection costs and improving adaptability.

**Optimization via Renoise Inversion.** In addition, to accurately estimate the trajectory preference distribution of the diffusion model, we employ the Renoise Inversion method for estimating the sampling trajectory. This approach replaces the forward-process-based estimation used in (Wallace et al., 2023), effectively addressing the issue of objective misalignment and facilitating more efficient fine-tuning.

Extensive experiments demonstrate the advantages of our contributions. We compare SmPO-Diffusion with state-of-the-art baselines for human preference alignment. Our approach outperforms existing methods across multiple human preference evaluation metrics (refer to Table 2) and demonstrates significant improvements in image generation quality (examples shown in Figures 1, 3 and 4) and training efficiency (26 × faster than Diffusion-KTO as per Table 1).

## 2 Related Works

**Text-to-Imgae Generative Models.** Early research utilizes GANs for T2I generation (Reed et al., 2016; Karras et al., 2018), while diffusion models (Ho et al., 2020; Nichol & Dhariwal, 2021) and flow matching (Liu et al., 2022; Lipman et al., 2022; Albergo & Vanden-Eijnden, 2022) have become the dominant approaches for image synthesis. Despite the ability of Stable Diffusion models (Rombach et al., 2021; Podell et al., 2023; Esser et al., 2024) to produce high-quality images, these models are generally trained on large, noisy datasets, which can lead to results that are contrary to human intent. Our work explores the effectiveness of using synthetically preference dataset (Kirstain et al., 2023) to improve pre-trained T2I generative models through techniques that learn from human/AI feedback .

**Preference Alignment of Diffusion Models.** Inspired by RLHF methos in LLMs, text-image reward models (Schuhmann et al., 2022; Radford et al., 2021a; Wu et al., 2023; Zhang et al., 2024a; Xu et al., 2023; Wu et al., 2024) have been increasingly developed and applied for fine-tuning

T2I generation models. With these reward models, DRaFT (Clark et al., 2023) and AlignProp (Prabhudesai et al., 2023) update the diffusion model's sampling trajectory using differentiable reward propagation, while DPOK (Fan et al., 2023) and DDPO (Black et al., 2023) regard the sampling process as a Markov decision process and apply policy gradient methods for training. However, these RL techniques require resource-intensive training objectives and are prone to reward hacking. To address these issues, Diffusion-DPO (Wallace et al., 2023) and D3PO (Yang et al., 2023) fine-tune the diffusion model for the unique global optimal policy using DPO techniques, while DDIM-InPO (Lu et al., 2025) uses DDIM Inversion to align specific latent variables, leading to a series of variants (Yang et al., 2024; Li et al., 2024a; Hong et al., 2024b; Lou et al., 2024; Gu et al., 2024). However, the flawed design of human preferences in the dataset often results in over-optimization and low training efficiency for these methods. Differently, our method investigates more accurate modeling and estimation of human preferences.

## 3  Preliminaries

**Diffusion Models.** Diffusion models (Nichol & Dhariwal, 2021) consist of a forward process $q(\boldsymbol{x}_{0:T})$ that progressively adds noise to data $\boldsymbol{x}_0 \sim q(\boldsymbol{x}_0)$, and a learned denoising process $p_\theta(\boldsymbol{x}_{0:T})$ to reconstruct the data from $p_\theta(\boldsymbol{x}_T) \sim \mathcal{N}(\boldsymbol{0}, \mathbf{I})$. The forward process is defined as a Markov process $q(\boldsymbol{x}_t|\boldsymbol{x}_{t-1}) = \mathcal{N}(\boldsymbol{x}_t; \sqrt{\alpha_t}\boldsymbol{x}_{t-1}, (1-\alpha_t)\mathbf{I})$, where $\alpha_t$ is the noise schedule. The denoising process takes the form of $p_\theta(\boldsymbol{x}_{t-1}|\boldsymbol{x}_t) = \mathcal{N}(\boldsymbol{x}_{t-1}; \mu_\theta^t(\boldsymbol{x}_t), \Sigma_t(\boldsymbol{x}_t))$. The training objective is to maximize the variational lower bound (VLB) associated with the parameterized denoiser $\boldsymbol{\epsilon}_\theta^t$:

$$\mathcal{L}_{\text{DM}} = \mathbb{E}_{x_0, x_t, t \sim \mathcal{U}(0,T), \boldsymbol{\epsilon} \sim \mathcal{N}(\boldsymbol{0}, \mathbf{I})}[\lambda(t) \left\| \boldsymbol{\epsilon}_\theta^t(\boldsymbol{x}_t) - \boldsymbol{\epsilon} \right\|_2^2] \tag{1}$$

where $\boldsymbol{x}_0 \sim q(\boldsymbol{x}_0), \boldsymbol{x}_t \sim q(\boldsymbol{x}_t|\boldsymbol{x}_0) = \mathcal{N}(\boldsymbol{x}_t; \sqrt{\bar{\alpha}_t}\boldsymbol{x}_0, (1-\bar{\alpha}_t)\mathbf{I})$. $\bar{\alpha}_t = \prod_{i=1}^t \alpha_t$ and $\lambda(t)$ is a pre-specified positive weighting function (Ho et al., 2020; Song et al., 2020b).

Denoising Diffusion Implicit Models (DDIM) (Song et al., 2020a) formulate the denoising process as an ODE, deterministically generating data. The discrete form of the process can be expressed as:

$$\boldsymbol{x}_t = \sqrt{\frac{\alpha_t}{\alpha_{t+1}}}\boldsymbol{x}_{t+1} + \left( \sqrt{\frac{1-\alpha_t}{\alpha_t}} - \sqrt{\frac{1-\alpha_{t+1}}{\alpha_{t+1}}} \right) \boldsymbol{\epsilon}_\theta^{t+1}(\boldsymbol{x}_{t+1}). \tag{2}$$

This proccess can also be reverse-integrated, starting from a noise-free image $\boldsymbol{x}_0$ to estimate $\boldsymbol{x}_t$ at any time step $t$, which is known as DDIM Inversion (Mokady et al., 2023).

**Direct Preference Optimization.** Given a predefined human preference dataset $\mathcal{D} = \{(\boldsymbol{c}, \boldsymbol{x}_0^w, \boldsymbol{x}_0^l)\}$, each sample consists of a prompt $\boldsymbol{c}$ and a pair of images generated by a reference model, with each pair labeled as the winner

$\boldsymbol{x}_0^w$ or loser $\boldsymbol{x}_0^l$ based on human preferences. The Bradley-Terry (BT) (Bradley & Terry, 1952) model defines pairwise preference as follows:

$$p_{\text{BT}}(\boldsymbol{x}_0^w \succ \boldsymbol{x}_0^l | \boldsymbol{c}) = \sigma(r(\boldsymbol{x}_0^w, \boldsymbol{c}) - r(\boldsymbol{x}_0^l, \boldsymbol{c})) \tag{3}$$

where $\sigma(\cdot)$ is the sigmoid function and $r(\boldsymbol{x}_0^*, \boldsymbol{c})$ is a reward model which can be parameterized by a neural network and trained with maximum likelihood estimation (MLE).

Building on this, (Rafailov et al., 2024) design the following DPO loss function and demonstrate its equivalence to a reinforcement learning (Sutton & Barto, 2018) process (such as PPO or REINFORCE) with an explicit reward model:

$$\mathcal{L}_{\text{DPO}} = -\mathbb{E}_{\substack{(\boldsymbol{x}_0^w, \boldsymbol{x}_0^l \\ , \boldsymbol{c}) \sim \mathcal{D}}} \log \sigma \left( \beta \log \frac{p_\theta(\boldsymbol{x}_0^w | \boldsymbol{c})}{p_{\text{ref}}(\boldsymbol{x}_0^w | \boldsymbol{c})} - \beta \log \frac{p_\theta(\boldsymbol{x}_0^l | \boldsymbol{c})}{p_{\text{ref}}(\boldsymbol{x}_0^l | \boldsymbol{c})} \right) \tag{4}$$

where $p_{\text{ref}}(\boldsymbol{x}_0^* | \boldsymbol{c})$ is the reference distribution and $\beta$ is a hyperparameter that governs regularization.

**DPO for diffusion models.** For diffusion models, since directly computing the image's probability distribution $p(\boldsymbol{x}_0^* | \boldsymbol{c})$ is not feasible, Wallace et al. (2023) propose a method for optimizing the upper bound of the original DPO objective function and derive a differentiable objective:

$$\mathcal{L}_{\text{DPO-Diffusion}}(\theta) = -\mathbb{E}_{(\boldsymbol{x}_0^w, \boldsymbol{x}_0^l, \boldsymbol{c}) \sim \mathcal{D}} \log \sigma$$
$$\left( \beta \mathbb{E}_{\substack{\boldsymbol{x}_{1:T}^w \sim p_\theta^{\boldsymbol{c}}(\boldsymbol{x}_{1:T}^w | \boldsymbol{x}_0^w) \\ \boldsymbol{x}_{1:T}^l \sim p_\theta^{\boldsymbol{c}}(\boldsymbol{x}_{1:T}^l | \boldsymbol{x}_0^l)}} \left[ \log \frac{p_\theta^{\boldsymbol{c}}(\boldsymbol{x}_{0:T}^w)}{p_{\text{ref}}^{\boldsymbol{c}}(\boldsymbol{x}_{0:T}^w)} - \log \frac{p_\theta^{\boldsymbol{c}}(\boldsymbol{x}_{0:T}^l)}{p_{\text{ref}}^{\boldsymbol{c}}(\boldsymbol{x}_{0:T}^l)} \right] \right) \tag{5}$$

where $p_\theta^{\boldsymbol{c}}(\cdot)$ represents $p_\theta(\cdot | \boldsymbol{c})$ for compactness. They estimate Equation (5) based on the following formula:

$$\mathcal{L}(\theta) = -\mathbb{E}_{\substack{(\boldsymbol{x}_0^w, \boldsymbol{x}_0^l, \boldsymbol{c}) \sim \mathcal{D}, \\ t \sim \mathcal{U}(0,T)}} \log \sigma(-\beta(\boldsymbol{s}_\theta^t(\boldsymbol{x}_0^w, \boldsymbol{c}) - \boldsymbol{s}_\theta^t(\boldsymbol{x}_0^l, \boldsymbol{c}))) \tag{6}$$

Here the score function $\boldsymbol{s}_\theta^t$ is defined as:

$$\boldsymbol{s}_\theta^t(\boldsymbol{x}_0^*, \boldsymbol{c}) = \|\boldsymbol{\epsilon}^* - \boldsymbol{\epsilon}_\theta^t(\boldsymbol{x}_t^*, \boldsymbol{c})\|_2^2 - \|\boldsymbol{\epsilon}^* - \boldsymbol{\epsilon}_{\text{ref}}^t(\boldsymbol{x}_t^*, \boldsymbol{c})\|_2^2. \tag{7}$$

where $\boldsymbol{\epsilon}^* \sim \mathcal{N}(\boldsymbol{0}, \mathbf{I})$ and $\boldsymbol{x}_t^* = \sqrt{\bar{\alpha}_t}\boldsymbol{x}_0^* + \sqrt{1-\bar{\alpha}_t}\boldsymbol{\epsilon}^*$.

## 4  Methodology

In this section, we introduce our SmPO-Diffusion for aligning diffusion models with human preference. *Our motivation is to enable more precise modeling and estimation of human preferences within the framework of diffusion preference optimization.* First, to account for the inherent variability in human preferences, we employ a smooth preference distribution for modeling (Section 4.1). Furthermore, we estimate the trajectory preference distribution of the diffusion model through Renoise Inversion (Section 4.2). The overview of the SmPO-Diffusion is illustrated in Figure 2.

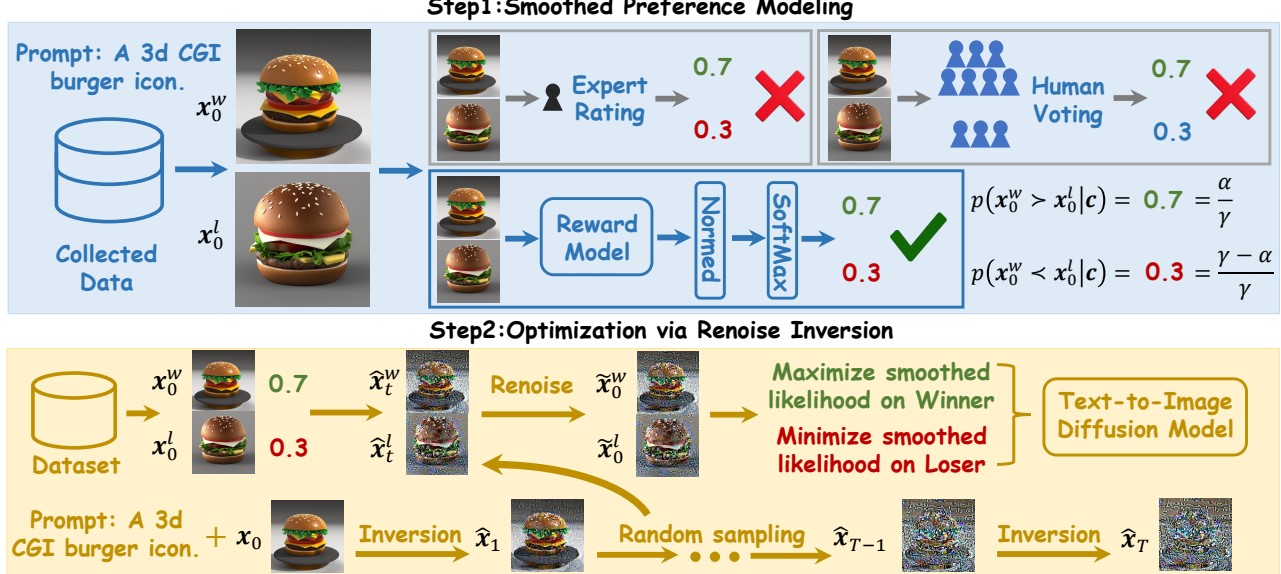

Figure 2: **Overview of SmPO-Diffusion.** We present two steps of SmPO-Diffusion: (1) Smoothed preference modeling. We calculate smoothed preference labels for all image pairs in the dataset. Unlike traditional methods such as expert ratings and human voting, we employ a reward model (PickScore) to assign scores to preference pairs. These reward scores are subsequently normalized and processed using a Softmax function to derive the final smoothed labels. (2) Optimization via Renoise Inversion. To estimate Equation (10), we use Renoise Inversion (repeat a few steps of Equation (13) for Inversion, and perform one step of Renoise according to Equation (14)) to estimate the diffusion model sampling trajectory, maximizing the smoothed log-likelihood of preferences to diffusion models. The final loss function is formulated as Equation (15).

## 4.1 Smoothed Preference Modeling

When sampling any pair of images from the dataset $\mathcal{D}$, individual preferences can exhibit significant variability. In this case, we employ a smooth distribution to model human preferences (Furuta et al., 2024). Assuming that the datasets of the winners $\boldsymbol{x}_0^w$ and losers $\boldsymbol{x}_0^l$ are sampled from a smoothed probability density $\tilde{p}(\cdot|\boldsymbol{c})$, we model the complex probability distribution using a weighted average:

$$
\begin{cases}
\tilde{p}(\boldsymbol{x}_0^w|\boldsymbol{c}) = \dfrac{p(\boldsymbol{x}_0^w|\boldsymbol{c})^\alpha p(\boldsymbol{x}_0^l|\boldsymbol{c})^{\gamma-\alpha}}{Z_{\boldsymbol{p}}^w(\boldsymbol{c})} \\[2ex]
\tilde{p}(\boldsymbol{x}_0^l|\boldsymbol{c}) = \dfrac{p(\boldsymbol{x}_0^w|\boldsymbol{c})^{\gamma-\alpha} p(\boldsymbol{x}_0^l|\boldsymbol{c})^{\alpha}}{Z_{\boldsymbol{p}}^l(\boldsymbol{c})}
\end{cases}
\tag{8}
$$

where factors $Z_{\boldsymbol{p}}^w(\boldsymbol{c}) = \sum_{\boldsymbol{x}_0^w,\boldsymbol{x}_0^l} p(\boldsymbol{x}_0^w|\boldsymbol{c})^\alpha p(\boldsymbol{x}_0^l|\boldsymbol{c})^{\gamma-\alpha}$ and $Z_{\boldsymbol{p}}^l(\boldsymbol{c}) = \sum_{\boldsymbol{x}_0^w,\boldsymbol{x}_0^l} p(\boldsymbol{x}_0^w|\boldsymbol{c})^{\gamma-\alpha} p(\boldsymbol{x}_0^l|\boldsymbol{c})^\alpha$ are for normalization. Since these values are challenging to estimate, the normalization term is approximated as a constant and omitted. Here, the weighting factor $\alpha$ balances the relative contributions of the winner and loser distributions, while the sensitivity factor $\gamma$ governs the sensitivity of the mixture distribution to variations in parameters. Weighted averaging serves as a smoothing technique, enabling effective modulation of the likelihood scale. Replacing $p(\cdot|\boldsymbol{c})$ in Equation (4)

with $\tilde{p}(\cdot|\boldsymbol{c})$, yields the following training objective:

$$
\begin{aligned}
\mathcal{L}_{\mathrm{SmPO}} &= -\mathbb{E}_{\substack{(\boldsymbol{x}_0^w,\boldsymbol{x}_0^l \\ ,\boldsymbol{c})\sim\mathcal{D}}} \log\sigma\left(\beta\log\frac{\tilde{p}_\theta^{\boldsymbol{c}}(\boldsymbol{x}_0^w)}{\tilde{p}_{\mathrm{ref}}^{\boldsymbol{c}}(\boldsymbol{x}_0^w)} - \beta\log\frac{\tilde{p}_\theta^{\boldsymbol{c}}(\boldsymbol{x}_0^l)}{\tilde{p}_{\mathrm{ref}}^{\boldsymbol{c}}(\boldsymbol{x}_0^l)}\right) \\
&= -\mathbb{E}_{\substack{(\boldsymbol{x}_0^w,\boldsymbol{x}_0^l \\ ,\boldsymbol{c})\sim\mathcal{D}}} \log\sigma\left((2\alpha-\gamma)\beta(\log\frac{p_\theta^{\boldsymbol{c}}(\boldsymbol{x}_0^w)}{p_{\mathrm{ref}}^{\boldsymbol{c}}(\boldsymbol{x}_0^w)} - \log\frac{p_\theta^{\boldsymbol{c}}(\boldsymbol{x}_0^l)}{p_{\mathrm{ref}}^{\boldsymbol{c}}(\boldsymbol{x}_0^l)})\right).
\end{aligned}
\tag{9}
$$

Equation (8) represents one of the options for regularization design. If we use binary labels, i.e., $\alpha = 1$, $\gamma = 1$, Equation (9) reduces to the original form Equation (4).

**Modeling for diffusion models.** Following (Wallace et al., 2023), we redistribute the rewards $r(\boldsymbol{x}_0, c)$ over all potential diffusion trajectories $p_\theta(\boldsymbol{x}_{1:T}|\boldsymbol{x}_0,\boldsymbol{c})$, with the goal of minimizing the KL-divergence between the joint probability distributions $\mathbb{D}_{\mathrm{KL}}(p_\theta(\boldsymbol{x}_{0:T}|\boldsymbol{c})||p_{\mathrm{ref}}(\boldsymbol{x}_{0:T}|\boldsymbol{c}))$. Thus, we arrive at the following objective for diffusion models:

$$
\mathcal{L}_{\mathrm{SmPO-Diffusion}} := -\mathbb{E}_{(\boldsymbol{x}_0^w,\boldsymbol{x}_0^l,\boldsymbol{c})\sim\mathcal{D}} \log\sigma\Big((2\alpha-\gamma)
$$
$$
\beta\mathbb{E}_{\substack{\boldsymbol{x}_{1:T}^w\sim p_\theta^{\boldsymbol{c}}(\boldsymbol{x}_{1:T}^w|\boldsymbol{x}_0^w) \\ \boldsymbol{x}_{1:T}^l\sim p_\theta^{\boldsymbol{c}}(\boldsymbol{x}_{1:T}^l|\boldsymbol{x}_0^l)}}\left[\log\frac{p_\theta^{\boldsymbol{c}}(\boldsymbol{x}_{0:T}^w)}{p_{\mathrm{ref}}^{\boldsymbol{c}}(\boldsymbol{x}_{0:T}^w)} - \log\frac{p_\theta^{\boldsymbol{c}}(\boldsymbol{x}_{0:T}^l)}{p_{\mathrm{ref}}^{\boldsymbol{c}}(\boldsymbol{x}_{0:T}^l)}\right]\Big).
\tag{10}
$$

When preferences are more similar, meaning humans find it harder to differentiate image quality, the loss function decreases further; otherwise, it increases. This modeling

approach more accurately reflects human preferences. To achieve this goal, how should $\alpha$ and $\gamma$ be formulated? In our work, we define the weight-to-sensitivity ratio $\alpha/\gamma$ as the probability of the winner $p(x_0^w \succ x_0^l | c)$, ensuring better alignment with human preferences.

**Smoothed Preference of Reward Model.** Inspired from prior works (Lee et al., 2023a; Black et al., 2023; Fan et al., 2023), we adopt the text-image reward model as a reliable alternative to expert rating or human voting. It is because AI reward models demonstrate strong consistency with human preferences, a concept commonly used in the RLHF literature. To produce smooth preference labels, reward scores are computed for all image pairs, with the higher-scoring image selected as the winner. Specifically, we feed the prompt $c$ and image $x_0^*$ into a reward model $r$ to derive reward scores $r(x_0^*, c)$. Given $\mathcal{D}$, we obtain a set of reward scores, denoted as $\mathcal{R}_\mathcal{D} = \{(r(x_0^w, c), r(x_0^l, c))\}_{(x_0^w, x_0^l, c) \sim \mathcal{D}}$. To achieve a balanced probability distribution, we normalize the reward score as described below:

$$r'(x_0^*, c) = \frac{r(x_0^*, c) - \max(\mathcal{R}_\mathcal{D})}{\max(\mathcal{R}_\mathcal{D}) - \min(\mathcal{R}_\mathcal{D})}. \quad (11)$$

For various preference data pairs, the weight-to-sensitivity ratio $\alpha/\gamma$ is modeled as:

$$\frac{\alpha}{\gamma}(x_0^w, x_0^l) = \frac{\exp(r'(x_0^w, c))}{\exp(r'(x_0^w, c)) + \exp(r'(x_0^l, c))}. \quad (12)$$

In our work, we employ PickScore (Kirstain et al., 2023) as the reward model. To control the effect of weighting factor $\alpha$ on the fluctuations of the loss function, we set sensitivity parameter $\gamma$ to a fixed value.

### 4.2 Optimization via Renoise Inversion

To perform optimization on Equation (10), we need to sample $x_{1:T} \sim p_\theta^c(x_{1:T}|x_0)$. Given that the sampling process is intractable, previous methods (Wallace et al., 2023; Hong et al., 2024b) replace the reverse process with the forward noisy process $q(x_{1:T}|x_0)$. Since the noise is randomly drawn from a Gaussian distribution, it results in inaccurate estimation of the loss function. We consider this to be the primary cause of the observed inefficiency in training.

In order to improve the estimation of Equation (10), we apply an Inversion technique to evaluate the diffusion sampling process $p_\theta^c(x_{1:T}|x_0)$. Given image $x_0$, to compute $x_t$ at any time step $t$, we first apply DDIM Inversion (Mokady et al., 2023) iteratively to derive an approximate estimate:

$$x_t = \sqrt{\frac{\alpha_t}{\alpha_{t-1}}} x_{t-1} + \left( \sqrt{\frac{1-\alpha_t}{\alpha_t}} - \sqrt{\frac{1-\alpha_{t-1}}{\alpha_{t-1}}} \right) \epsilon_\theta^{t-1}(x_{t-1}, c). \quad (13)$$

Here, we denote the estimate of $x_t$ obtained as $\hat{x}_t$. However, Equation (13) assumes that a sufficient number of inversion

steps are taken, which results in a time-intensive inversion process that is detrimental to improving training efficiency. Therefore, inspired from (Garibi et al., 2024), we employ a few-step DDIM Inversion (fewer than 10 steps), followed by an additional single ReNoise step:

$$\tilde{x}_t = \sqrt{\frac{\alpha_t}{\alpha_{t-1}}} \hat{x}_{t-1} + \left( \sqrt{\frac{1-\alpha_t}{\alpha_t}} - \sqrt{\frac{1-\alpha_{t-1}}{\alpha_{t-1}}} \right) \epsilon_\theta^t(\hat{x}_t, c). \quad (14)$$

Following (Wallace et al., 2023), once we obtain $\tilde{x}_t$ and we can estimate Equation (10) in the following manner:

$$\mathcal{L}(\theta) = -\mathbb{E}_{t,\mathcal{D}} \log \sigma(-(2\alpha - \gamma)\beta(\tilde{s}_\theta^t(x_0^w, c) - \tilde{s}_\theta^t(x_0^l, c))) \quad (15)$$

where score function is defined as:

$$\tilde{s}_\theta^t(x_0^*, c) = \|\tau_t^* - \epsilon_\theta^t(\tilde{x}_t^*, c)\|_2^2 - \|\tau_t^* - \epsilon_{\text{ref}}^t(\tilde{x}_t^*, c)\|_2^2 \quad (16)$$

where $\tau_t^* = (\tilde{x}_t^* - \sqrt{\bar{\alpha}_t} x_0^*)/\sqrt{1-\bar{\alpha}_t}$. This loss function drives denoising $x_0^w$ at point $\tilde{x}_t^w$ to improve more significantly than denoising denoising $x_0^l$ at point $\tilde{x}_t^l$. Unlike the random noise addition process in Diffusion-DPO, this approach enables fine-tuning variables highly correlated with the image, thereby enhancing training efficiency.

## 5 Experiments

### 5.1 Setup

**Implementation Details.** Our method is applied to align the Stable Diffusion 1.5 (SD1.5) and Stable Diffusion XL-base-1.0 (SDXL). We utilize the Pick-a-Pic v2 dataset, which contains 851,293 data pairs and 58,960 unique text prompts as (Wallace et al., 2023). We employ AdamW (Loshchilov, 2017) as the optimizer for SD1.5 and Adafactor (Shazeer & Stern, 2018) for SDXL. All experiments are performed on 8 A800 GPUs, with each GPU handling a batch size of 1 data pair. Through 128 gradient accumulation steps, an batch size of 1024 data pairs is achieved. The learning rate is set to $\frac{2000}{\beta} 2.048^{-8}$, with a linear warm-up phase. For SD1.5, $\beta$ is set to 2000, while for SDXL, $\beta$ is set to 5000.

**Evaluation.** We compare SmPO-Diffusion with existing baselines across three dimensions: automatic preference metrics, user studies, and training resource consumption. In this work, we compare the following baseline methods: Supervised Fine-Tuning (SFT), Diffusion-DPO, Diffusion-KTO (for SD1.5), and MaPO (for SDXL). These baselines share a common focus on human-aligned image generation but differ fundamentally in their technical mechanisms: Diffusion-KTO adopts the Kahneman-Tversky model to represent human utility instead of maximizing the log-likelihood of preferences. MaPO jointly maximizes the likelihood margin between the preferred and dispreferred datasets and the likelihood of the preferred sets, learning preference without reference model. To ensure a fair comparison, all models are fine-tuned on the Pick-a-Pic v2

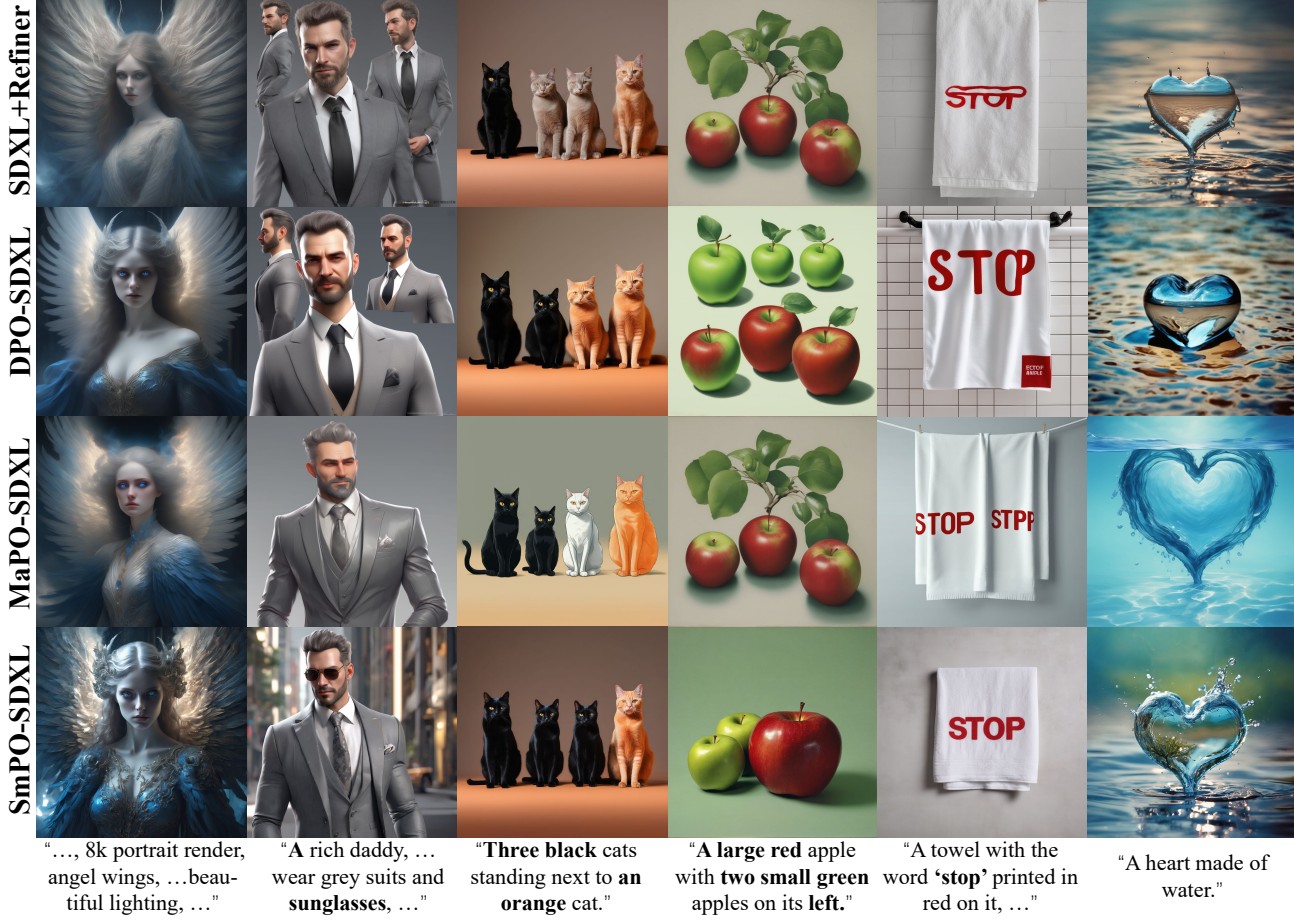

**Figure 3: Qualitative comparison.** We provide a qualitative comparison of SmPO-Diffusion and other different preference optimization methods (Refiner, Diffusion-DPO, MaPO) for SDXL. The results indicate that our model generates the highest quality images, including aspects such as llumination and spatial composition, text rendering, visual appeal, and others.

dataset. To comprehensively assess quality of image generation, we use five reward evaluators: CLIP (Radford et al., 2021a) for text-image alignment, LAION aesthetic classifier and Imagereward (Xu et al., 2023) for image quality, PickScore (Kirstain et al., 2023) and HPSv2.1 (Wu et al., 2023) for simulating human preferences. During testing, images are generated using the Parti-Prompts (1632 prompts) (Yu et al., 2022) and HPDv2 (3200 prompts) (Wu et al., 2023) text test sets with the same seed. We use the median of reward scores and the win-rate as automatic preference metrics, where the win-rate indicates the frequency with which the reward evaluator prefers images from SmPO-Diffusion over those from the baselines. Additionally, we compare the GPU hours required for training across different methods.

### 5.2 Primary Results

**Qualitative results.** Figures 1 and 3 present the qualitative comparison results of our SmPO-Diffusion and other baselines used for fine-tuning base SDXL model. First, as

Table 1: **Computational cost comparison.** We report the NVIDIA A800 GPU hours required for training our SmPO-Diffusion and the baselines on SDXL and SD1.5.

| Model | GPU Hours ↓ | Model | GPU Hours ↓ |
|---|---|---|---|
| DPO-SDXL | ∼ 976.0 | DPO-SD1.5 | ∼ 204.8 |
| MaPO-SDXL | ∼ 834.4 | KTO-SD1.5 | ∼ 1056.0 |
| SmPO-SDXL | ∼ **150.8** | SmPO-SD1.5 | ∼ **41.3** |

illustrated in Figure 1, SmPO-SDXL improves the quality of generated images while achieving better alignment with human preferences, and it also resolves some failure cases. Additionally, as demonstrated in Figure 3, compared to other advanced preference optimization methods for SDXL (Refiner, DPO-SDXL, MaPO-SDXL), our SmPO-SDXL produces images of the highest quality, with notable enhancements in illumination, spatial composition, text rendering, visual appeal, and other key aspects. These hidden advantages lie within human preferences, further validating the effectiveness of our method.

Table 2: **Quantitive comparison.** We employ the HPDv2 and Parti-Prompts test sets to generate images using each SDXL and SD1.5 model. By performing automatic evaluation of each reward evaluator, we report the median reward score (Score) of the images generated by all models and show the win-rate (%) of our SmPO-Diffusion compared to the corresponding baseline models (WR). Higher scores and win-rates demonstrate the superiority of our method. In the Score column, the highest value is displayed in **bold**, the second highest is underlined. Additionally, win-rates exceeding 50 % are highlighted.

| Model | HPDv2 (3200 prompts) | | | | | | | | | | Parti-Prompts (1632 prompts) | | | | | | | | | |
| --- | --- | --- | --- | --- | --- | --- | --- | --- | --- | --- | --- | --- | --- | --- | --- | --- | --- | --- | --- | --- |
| | PickScore↑ | | HPSv2.1↑ | | ImReward↑ | | Aesthetic↑ | | CLIP↑ | | PickScore↑ | | HPSv2.1↑ | | ImReward↑ | | Aesthetic↑ | | CLIP↑ | |
| | Score | WR | Score | WR | Score | WR | Score | WR | Score | WR | Score | WR | Score | WR | Score | WR | Score | WR | Score | WR |
| SDXL | 22.75 | 88.1 | 28.45 | 94.5 | 0.881 | 79.3 | 6.114 | 64.7 | 38.36 | 56.2 | 22.63 | 86.2 | 27.45 | 92.6 | 0.929 | 81.9 | 5.753 | 74.0 | 35.53 | 55.6 |
| SFT | 22.17 | 95.4 | 28.39 | 90.3 | 0.756 | 79.5 | 5.989 | 72.8 | 37.66 | 62.9 | 21.98 | 95.2 | 27.00 | 91.1 | 0.660 | 83.7 | 5.705 | 74.2 | 34.77 | 64.1 |
| DPO | 23.13 | 77.2 | 30.06 | 86.7 | 1.184 | 62.9 | 6.112 | 67.7 | 38.86 | 50.5 | 22.93 | 77.2 | 28.89 | 82.5 | 1.280 | 67.8 | 5.813 | 70.6 | 36.30 | 51.5 |
| MaPO | 22.81 | 86.3 | 29.11 | 90.8 | 1.224 | 60.5 | 6.309 | 47.6 | 38.17 | 58.1 | 22.83 | 80.2 | 28.47 | 84.9 | 1.269 | 63.5 | 6.012 | 47.3 | 35.24 | 60.7 |
| SmPO | 23.62 | - | 32.53 | - | 1.331 | - | 6.264 | - | 38.88 | - | 23.35 | - | 30.73 | - | 1.429 | - | 5.959 | - | 36.34 | - |
| SD1.5 | 20.83 | 88.0 | 23.61 | 93.9 | -0.078 | 85.9 | 5.390 | 81.5 | 34.71 | 69.3 | 21.40 | 74.5 | 24.97 | 89.2 | 0.121 | 77.3 | 5.355 | 76.4 | 33.11 | 63.1 |
| SFT | 21.64 | 75.8 | 28.60 | 65.7 | 0.738 | 62.1 | 5.725 | 63.3 | 36.24 | 61.4 | 21.77 | 65.6 | 28.17 | 62.3 | 0.702 | 59.8 | 5.592 | 55.9 | 34.07 | 59.5 |
| DPO | 21.29 | 79.1 | 25.11 | 89.5 | 0.195 | 80.1 | 5.530 | 72.7 | 35.58 | 64.6 | 21.62 | 66.5 | 25.79 | 84.9 | 0.386 | 71.2 | 5.442 | 70.2 | 33.54 | 59.7 |
| KTO | 21.54 | 73.4 | 28.28 | 66.8 | 0.706 | 61.9 | 5.692 | 65.1 | 35.92 | 64.5 | 21.77 | 65.8 | 28.05 | 64.5 | 0.677 | 59.9 | 5.547 | 62.8 | 34.11 | 59.0 |
| SmPO | 22.08 | - | 29.31 | - | 0.885 | - | 5.831 | - | 37.19 | - | 22.00 | - | 28.81 | - | 0.911 | - | 5.636 | - | 34.72 | - |

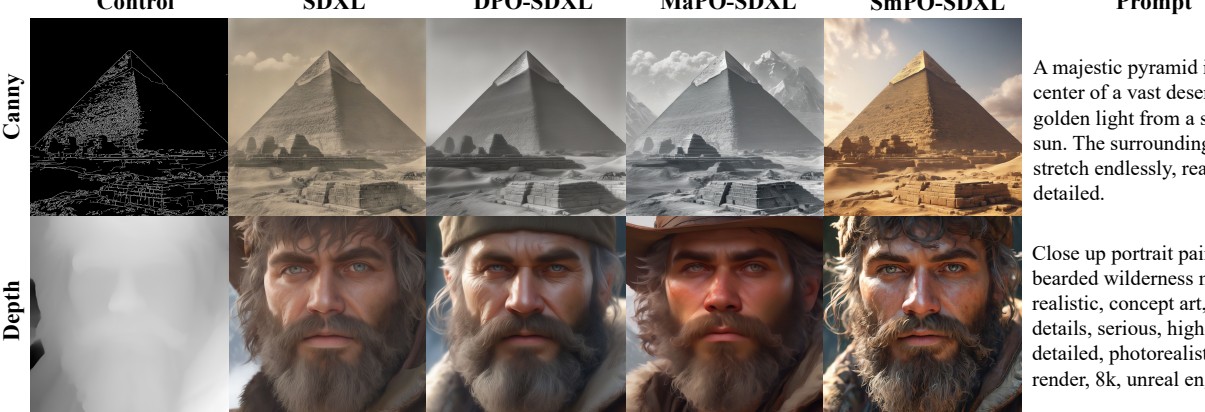

Figure 4: **Conditional generation comparison.** SmPO-SDXL produces images with the highest quality (cf. Section 5.3).

**Computational cost.** As illustrated in Table 1, our SmPO-Diffusion requires around 150.8 and 41.3 A800 GPU hours to train SDXL and SD1.5, respectively. These include 5.3 A800 GPU hours of PickScore reward model training. In comparison, fine-tuning SDXL and SD1.5 with Diffusion-DPO demands 976 and 204.8 GPU hours, respectively. The results demonstrate that SmPO-SDXL and SmPO-SD1.5 achieve significant improvements in generation quality while reducing training time to only 15.5% and 20.2% of that required by DPO-SDXL and DPO-SD1.5, respectively. Compared to the latest methods, SmPO-SDXL requires only 18.1% of the training time of MaPO-SDXL, and SmPO-SD1.5 requires merely 3.9% of the training time of KTO-SD1.5, resulting in a 26-fold efficiency improvement. We attribute this remarkable efficiency gain to the more precise modeling and estimation of human preferences,

consistent with our motivation.

**Quantitative results.** Table 2 presents the reward scores of our SmPO-aligned diffusion models and baseline models, along with the win-rates of our model compared to baselines. Overall, our SmPO fine-tuned SDXL and SD1.5 models demonstrate superior performance over the baseline models on nearly all test datasets and reward evaluation metrics. For instance, on the HPDv2 test set, the median HPSv2.1 scores for SmPO-SDXL and SmPO-SD1.5 reach 32.53 and 29.31, respectively, reflecting significant improvements of +4.08 and +4.7 compared to Base-SDXL and Base-SD1.5. When compared to state-of-the-art baselines, the HPSv2.1 metric reveals that SmPO-SDXL achieves an 86.7% win-rate against DPO-SDXL on the HPDv2 test set, while SmPO-SD1.5 achieves a 66.8% win-rate against KTO-

Table 3: Enhancements from the proposed method modules.

|            | PickScore↑ | HPS↑  | ImReward↑ | Aesthe↑ | CLIP↑ |
|------------|-----------|-------|-----------|---------|-------|
| DPO        | 21.29     | 25.11 | 0.1947    | 5.530   | 35.58 |
| +Inversion | 21.72     | 28.71 | 0.7608    | 5.748   | 36.41 |
| +Renoise   | 21.87     | 29.01 | 0.7778    | 5.786   | 36.73 |
| +Smoothed  | **22.08** | **29.31** | **0.8854** | **5.831** | **37.19** |

Table 4: Impact of maximum DDIM Inversion steps.

| InvStep | PickScore↑ | HPS↑  | ImReward↑ | Aesthe↑ | CLIP↑ | G-H↓ |
|---------|-----------|-------|-----------|---------|-------|------|
| 2       | 22.06     | 29.22 | 0.8738    | 5.828   | 37.01 | ∼ **27.8** |
| 4       | 22.06     | 29.23 | 0.8764    | 5.822   | 36.92 | ∼ 32.3 |
| 9       | 22.08     | 29.31 | 0.8854    | 5.831   | **37.19** | ∼ 41.3 |
| 19      | **22.11** | **29.50** | **0.8964** | **5.844** | 37.13 | ∼ 67.3 |

Table 5: Impact of sensitivity factor $\gamma$ and regularization $\beta$.

| $(\gamma, \beta)$ | PickScore↑ | HPS↑  | ImReward↑ | Aesthe↑ | CLIP↑ |
|-------------|-----------|-------|-----------|---------|-------|
| (2,2000)    | 21.71     | 28.50 | 0.7746    | 5.712   | 36.41 |
| (5,2000)    | 21.94     | 28.57 | 0.7716    | 5.812   | 36.79 |
| (20,2000)   | 21.96     | 29.27 | 0.8542    | 5.798   | 36.93 |
| (10,2000)   | **22.08** | **29.31** | **0.8854** | **5.831** | **37.19** |
| (10,1000)   | 21.98     | 28.69 | 0.8313    | 5.823   | 36.97 |
| (10,3000)   | 21.95     | 29.13 | 0.8504    | 5.806   | 37.06 |
| (10,5000)   | 21.90     | 29.12 | 0.8312    | 5.795   | 36.86 |

Table 6: Impact of CFG during DDIM Inversion.

| InvCFG | PickScore↑ | HPS↑  | ImReward↑ | Aesthe↑ | CLIP↑ | G-H↓ |
|--------|-----------|-------|-----------|---------|-------|------|
| -7.5   | 21.82     | 29.06 | 0.7772    | 5.766   | 36.51 | ∼ 70.3 |
| -5     | 21.89     | 29.12 | 0.8047    | 5.776   | 36.77 | ∼ 70.3 |
| -1     | 21.79     | 29.27 | 0.8310    | 5.753   | 36.62 | ∼ 70.3 |
| 0      | 21.78     | 29.22 | 0.8311    | 5.734   | 36.51 | ∼ **41.3** |
| 1      | **22.08** | **29.31** | **0.8854** | **5.831** | **37.19** | ∼ **41.3** |
| 5      | 21.83     | 29.12 | 0.7972    | 5.741   | 36.44 | ∼ 70.3 |
| 7.5    | 21.81     | 29.01 | 0.7931    | 5.752   | 36.38 | ∼ 70.3 |

SD1.5. Similar trends are observed across other metrics, further substantiating the superiority of our approach.

## 5.3 Conditional generation results

Without the need for further training, the model can be directly applied to conditional generation tasks. We employ ControlNet to provide conditional control, which has been jointly pre-trained with the Base-SDXL model. As shown in Figure 4, we use canny map and depth map as additional conditions to control the T2I generation process. Experimental results demonstrate that the generated images successfully retain the strengths of the SmPO-aligned model.

## 5.4 Ablation Studies and Analysis

Tables 3 to 7 present the results of our ablation experiments on SD1.5. We report the median score of the reward evaluators on the HPDv2 test set across different experimental configurations, along with an analysis of the results.

**Proposed Method Enhancements.** The core of our method lies in estimating the diffusion sampling process using Renoise Inversion and employing a smooth preference modeling approach. As demonstrated in Table 3, estimating the latent variable $x_t$ via DDIM Inversion, followed by correcting $x_t$ with an additional step of Renoise, significantly improves the model's performance. This highlights the critical importance of accurately estimating the diffusion sampling process for diffusion models alignment. Furthermore, the integration of varied human preferences through smooth preference modeling substantially improves the quality of T2I generation. These two components comprehensively validate the effectiveness of our proposed method.

**Parameter Selections.** Table 4 illustrates the influence of the maximum DDIM Inversion steps required on results.

Table 7: Impact of weight-to-sensitivity ratio using different reward models. We report the performance comparison where rows represent different training signals and columns denote evaluation metrics. Upward arrows (↑) in column headers indicate higher values are preferable. The highest value is displayed in **bold**, the second highest is underlined.

| Reward    | PickScore↑ | HPSv2.1↑ | ImReward↑ | Aesthe↑ | CLIP↑ |
|-----------|-----------|----------|-----------|---------|-------|
| PickScore | **22.08** | 29.31 | 0.8854 | 5.831 | **37.19** |
| HPSv2.1   | 21.87     | **29.50** | **0.8891** | 5.790   | 36.81 |
| ImReward  | 21.77     | 28.98    | 0.8638    | 5.766   | 36.71 |
| Aesth     | 22.01 | 29.26 | 0.8321    | **5.958** | 36.16 |
| CLIP      | 21.78     | 28.77    | 0.8428    | 5.759   | 37.08 |

Experimental results indicate that configuring the step count to 19 achieves optimal image quality. However, this configuration substantially escalates the demand for training resources. To balance quality and efficiency, we fix the inversion steps at 9. Furthermore, classifier-free guidance (CFG) also plays a significant role in determining the efficacy of the inversion process. Mokady et al. (2023) point out that DDIM inversion is sensitive to the prompt $c$. Table 6 reports the influence of CFG during DDIM Inversion on the results, with experimental results demonstrate that optimal performance is achieved when CFG=1. Additionally, we set the sensitivity parameter $\gamma$ to 10. As shown in rows 2 to 5 of Table 5, if $\gamma$ is too small, the model is insensitive to changes in the reward score, resulting in suboptimal optimization. Conversely, setting $\gamma$ too high may cause over-optimization of the reward model. Moreover, we set $\beta$ to 2000. Rows 5 to 8 of Table 5 reveal that if $\beta$ is set too low, the diffusion model degenerates into a pure reward scoring model, while if set too high, the KL divergence penalty term overly restricts the model's flexibility during the adjustment process.

**Choice of Reward Model.** As shown in Table 7, training with text-aware preference estimation models, such as

PickScore and HPSv2.1, enhances both the visual appeal and the text rendering capabilities of images. However, other reward models prioritize enhancing specific performance metrics, such as aesthetic optimization, which improves the corresponding model capabilities but sacrifice some of the text rendering ability. We note that PickScore can be viewed as pseudo-labels from the Pick-a-Pick dataset, equivalent to cleaned data, making it more suitable as a reward model.

## 6   Conclusion and Discussion

In our work, recognizing the variability of human preferences, we introduce SmPO-Diffusion, a novel method for modeling preference distributions with a numerical upper bound estimation for optimizing the diffusion DPO objective. Initially, we replace the binary distribution with a smooth preference distribution modeled using a reward model. Additionally, we apply Renoise Inversion to estimate the trajectory preference distribution. Experimental results demonstrate that our method achieves SOTA performance on various human preference evaluation tasks while significantly reducing consumption of training resources. We posit that integrating preference alignment into online learning represents a promising future direction, enabling models to undergo continuous performance enhancement.

**Limitations.**   In this paper, we utilize the Pick-a-Pic v2 dataset for model training and evaluation. While this dataset provides a diverse range of image-text pairs, it is important to acknowledge that it may contain harmful or biased content, which could inadvertently influence our model's outputs. Specifically, we observe that certain societal biases within the dataset—such as gendered stereotypes—may lead the model to generate overly feminized representations in response to neutral or non-gendered prompts. This potential bias necessitates careful consideration during both training and inference to mitigate unintended consequences.

## Acknowledgments

This work was supported by the National Major Science and Technology Projects (the grant number 2022ZD0117000) and the National Natural Science Foundation of China (grant number 62202426).

## Impact Statement

This paper presents work whose goal is to advance the field of Machine Learning. While sharing the same abuse potential as conventional preference alignment methods - where malicious actors could train harmful T2I models by aligning with unethical preferences - our approach's negative effects can be substantially mitigated through rigorous dataset auditing and preference source monitoring.

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

# Appendix

## A   Background

**Conditional Generative Models.**   Diffusion models represent a category of generative models that create data by inverting a noise-introducing forward process. During the training phase, the neural network learns to approximate the reversal of this forward process, which systematically corrupts data with noise. By harnessing the generalization and approximation abilities of neural networks, diffusion models (Nichol & Dhariwal, 2021; Ho et al., 2020) are capable of generating diverse data samples that align with the distribution of the training dataset. These models can be broadly categorized into two main paradigms: denoising diffusion (Ho et al., 2020) and score-based matching (Song & Ermon, 2019; Song et al., 2020b), each providing unique theoretical frameworks and computational strategies for the generation process. In recent years, diffusion models and their variants, including Rectified Flow (Liu et al., 2022), have risen to prominence as the leading framework in generative modeling, showcasing exceptional performance in both output quality and training stability compared to earlier methods. Thanks to advancements in diffusion models (Peebles & Xie, 2023; Bao et al., 2023), they have also achieved remarkable progress in areas such as conditional image generation (Chen et al., 2023b; 2025), audio synthesis (Zhang et al., 2023a), and video creation (Bao et al., 2024; Blattmann et al., 2023a; Brooks et al., 2024; Blattmann et al., 2023b). In this study, we concentrate primarily on conditional image generation (Huang et al., 2024; Meng et al., 2021). A common strategy for generating desired images involves the use of text prompts (Hertz et al., 2022) as guidance. Textual information is typically converted into text embeddings using a pre-trained text encoder (Radford et al., 2021b), enabling text-to-image models to achieve impressive results. However, relying solely on textual input often results in limited precision. To improve image control with additional conditions, such as sketches, the prevailing approach involves training a dedicated control network and integrating it with the generative model. A prominent example is ControlNet (Zhang et al., 2023b), which trains an independent control network for each conditional image, fostering the development of various control techniques. In our work, we utilize edge maps and depth maps to guide the text-to-image model.

**Preference Alignment of Large Language Models.**   Reinforcement Learning from Human Feedback (RLHF) (Christiano et al., 2017) is a pivotal technique for aligning large language models (LLMs) with human preferences (Ouyang et al., 2022). This methodology typically involves two main steps: first, training a reward model to approximate human preferences, and second, applying reinforcement learning to optimize the policy, aiming to maximize the reward feedback received by the model. Leading LLMs, such as ChatGPT, have effectively integrated this fine-tuning approach. A central algorithm in reinforcement learning, Proximal Policy Optimization (PPO) (Schulman et al., 2017), requires the concurrent loading of multiple models, including the training model, reference model, critic model, and reward model. The substantial computational overhead and intricate training objectives associated with PPO often make its practical optimization challenging (Ouyang et al., 2022). To address these computational demands and improve training efficiency, Ahmadian et al. (2024) has shown that REINFORCE-style methods and their variants are highly effective within the RLHF framework. Additionally, alternative methods circumvent traditional reinforcement learning by using the reward model to rank prompt samples from LLMs and then collecting preference data for fine-tuning. For example, RAFT (Dong et al., 2023) focuses on supervised fine-tuning of high-reward samples, RRHF (Yuan et al., 2023) uses ranking loss for alignment, and Liu et al. (2023) applies rejection sampling optimization to gather preference data from the optimal policy.

Direct Preference Optimization (DPO) (Rafailov et al., 2024) eliminates the need for an explicit reward model by directly optimizing the policy, implicitly refining the reward scores embedded in the Bradley-Terry (BT) model. Similarly, IPO (Azar et al., 2024) argues that pairwise preferences cannot be replaced by pointwise rewards and provides a method for direct optimization based on preference probabilities. Furthermore, ORPO (Hong et al., 2024a) removes the requirement for a reference model by simultaneously performing supervised fine-tuning and preference optimization. In alternative reward configurations, KTO (Ethayarajh et al., 2024) adopts the Kahneman-Tversky model to represent human utility instead of maximizing the log-likelihood of preferences, while PRO (Song et al., 2024) utilizes reward ranking information to optimize the large language models .

**Further Diffusion Models Alignment.**   Beyond preference alignment in text-to-image diffusion models, other generative domains, such as video and 3D, have also adopted alignment methods customized for their unique data characteristics. For example, InstructVideo (Yuan et al., 2024) enhances text-to-video diffusion models by incorporating reward fine-tuning supplemented with human feedback. This method utilizes an image reward model to improve video quality while reducing fine-tuning costs through partial DDIM sampling. Similarly, Prabhudesai et al. (2024) aligns the base video diffusion model

using gradients from a publicly available pre-trained visual reward model. In the 3D domain, DreamReward (Ye et al., 2025) creates a text-based 3D dataset to train a reward model (Reward3D) and integrates the reward model's gradients into the score distillation sampling (Poole et al., 2022) framework. The alignment of human preferences in diffusion models is still in its early stages, with future advancements potentially including the adaptation of Large Language Model (LLM) alignment techniques to diffusion models and the extension of alignment approaches to additional modalities, such as audio and tactile feedback.

## B   Details of the Primary Derivation

In this section of the paper, we provide the full derivation of Equation (9):

$$
\begin{aligned}
\mathcal{L}_{\mathrm{SmPO}} &= -\mathbb{E}_{(\boldsymbol{x}_0^w,\boldsymbol{x}_0^l,\boldsymbol{c})\sim\mathcal{D}} \log\sigma\left(\beta\log\frac{\tilde{p}_\theta(\boldsymbol{x}_0^w|\boldsymbol{c})}{\tilde{p}_{\mathrm{ref}}(\boldsymbol{x}_0^w|\boldsymbol{c})} - \beta\log\frac{\tilde{p}_\theta(\boldsymbol{x}_0^l|\boldsymbol{c})}{\tilde{p}_{\mathrm{ref}}(\boldsymbol{x}_0^l|\boldsymbol{c})}\right) \\
&= -\mathbb{E}_{(\boldsymbol{x}_0^w,\boldsymbol{x}_0^l,\boldsymbol{c})\sim\mathcal{D}} \log\sigma\left(\beta\log\frac{\tilde{p}_\theta(\boldsymbol{x}_0^w|\boldsymbol{c})\tilde{p}_{\mathrm{ref}}(\boldsymbol{x}_0^l|\boldsymbol{c})}{\tilde{p}_{\mathrm{ref}}(\boldsymbol{x}_0^w|\boldsymbol{c})\tilde{p}_\theta(\boldsymbol{x}_0^l|\boldsymbol{c})}\right) \\
&= -\mathbb{E}_{(\boldsymbol{x}_0^w,\boldsymbol{x}_0^l,\boldsymbol{c})\sim\mathcal{D}} \log\sigma\left(\beta\log\frac{\frac{p_\theta(\boldsymbol{x}_0^w|\boldsymbol{c})^\alpha p_\theta(\boldsymbol{x}_0^l|\boldsymbol{c})^{\gamma-\alpha}}{Z_{\boldsymbol{p}_\theta}^w(\boldsymbol{c})}\frac{p_{\mathrm{ref}}(\boldsymbol{x}_0^w|\boldsymbol{c})^{\gamma-\alpha}p_{\mathrm{ref}}(\boldsymbol{x}_0^l|\boldsymbol{c})^\alpha}{Z_{\boldsymbol{p}_{\mathrm{ref}}}^w(\boldsymbol{c})}}{\frac{p_{\mathrm{ref}}(\boldsymbol{x}_0^w|\boldsymbol{c})^\alpha p_{\mathrm{ref}}(\boldsymbol{x}_0^l|\boldsymbol{c})^{\gamma-\alpha}}{Z_{\boldsymbol{p}_{\mathrm{ref}}}^w(\boldsymbol{c})}\frac{p_\theta(\boldsymbol{x}_0^w|\boldsymbol{c})^{\gamma-\alpha}p_\theta(\boldsymbol{x}_0^l|\boldsymbol{c})^\alpha}{Z_{\boldsymbol{p}_\theta}^w(\boldsymbol{c})}}\right) \\
&= -\mathbb{E}_{(\boldsymbol{x}_0^w,\boldsymbol{x}_0^l,\boldsymbol{c})\sim\mathcal{D}} \log\sigma\left(\beta\log\frac{p_\theta(\boldsymbol{x}_0^w|\boldsymbol{c})^\alpha p_\theta(\boldsymbol{x}_0^l|\boldsymbol{c})^{\gamma-\alpha}p_{\mathrm{ref}}(\boldsymbol{x}_0^w|\boldsymbol{c})^{\gamma-\alpha}p_{\mathrm{ref}}(\boldsymbol{x}_0^l|\boldsymbol{c})^\alpha}{p_{\mathrm{ref}}(\boldsymbol{x}_0^w|\boldsymbol{c})^\alpha p_{\mathrm{ref}}(\boldsymbol{x}_0^l|\boldsymbol{c})^{\gamma-\alpha}p_\theta(\boldsymbol{x}_0^w|\boldsymbol{c})^{\gamma-\alpha}p_\theta(\boldsymbol{x}_0^l|\boldsymbol{c})^\alpha}\right) \\
&= -\mathbb{E}_{(\boldsymbol{x}_0^w,\boldsymbol{x}_0^l,\boldsymbol{c})\sim\mathcal{D}} \log\sigma\left(\beta\log\frac{p_\theta(\boldsymbol{x}_0^w|\boldsymbol{c})^{2\alpha-\gamma}p_{\mathrm{ref}}(\boldsymbol{x}_0^l|\boldsymbol{c})^{2\alpha-\gamma}}{p_{\mathrm{ref}}(\boldsymbol{x}_0^w|\boldsymbol{c})^{2\alpha-\gamma}p_\theta(\boldsymbol{x}_0^l|\boldsymbol{c})^{2\alpha-\gamma}}\right) \\
&= -\mathbb{E}_{(\boldsymbol{x}_0^w,\boldsymbol{x}_0^l,\boldsymbol{c})\sim\mathcal{D}} \log\sigma\left((2\alpha-\gamma)\beta\log\frac{p_\theta(\boldsymbol{x}_0^w|\boldsymbol{c})p_{\mathrm{ref}}(\boldsymbol{x}_0^l|\boldsymbol{c})}{p_{\mathrm{ref}}(\boldsymbol{x}_0^w|\boldsymbol{c})p_\theta(\boldsymbol{x}_0^l|\boldsymbol{c})}\right) \\
&= -\mathbb{E}_{(\boldsymbol{x}_0^w,\boldsymbol{x}_0^l,\boldsymbol{c})\sim\mathcal{D}} \log\sigma\left((2\alpha-\gamma)\beta(\log\frac{p_\theta(\boldsymbol{x}_0^w|\boldsymbol{c})}{p_{\mathrm{ref}}(\boldsymbol{x}_0^w|\boldsymbol{c})} - \log\frac{p_\theta(\boldsymbol{x}_0^l|\boldsymbol{c})}{p_{\mathrm{ref}}(\boldsymbol{x}_0^l|\boldsymbol{c})})\right).
\end{aligned}
\tag{17}
$$

In the main body, $p_\theta^{\boldsymbol{c}}(\cdot)$ represents $p_\theta(\cdot|\boldsymbol{c})$ for compactness.

## C   Further Discussion

Human preferences are influenced by a variety of factors, including culture and geographical location, leading to significant variability among individuals. Consequently, our motivation is both well-founded and explicitly defined. In our experiments, we utilize the Pick-a-Pic v2 dataset, which features image quality ranging between SD1.5 and SDXL standards. The quantitative results (refer to Table 2) demonstrate that our method outperforms supervised fine-tuning and other preference optimization methods across both models, exhibiting superior adaptability. Our approach employs an offline fine-tuning methodology that relies on pre-existing datasets. However, it can be seamlessly transitioned to an online learning framework, enhancing its practical applicability.

## D   Experiment Details

**Additional Implementation details.**   During the text-to-image generation evaluation, the CFG (Classifier-Free Guidance) (Ho & Salimans, 2022) values for SD1.5 and SDXL are set to 7.5 and 5, respectively, following widely accepted standards. For tasks involving depth maps and Canny edges, the controlnet conditioning scales are set to 0.5 and 0.3, respectively, with CFG fixed at 5 for both. The random seeds for all comparative experiments are fixed as 0 to ensure reproducibility.

**Explanation of Chart.**   To facilitate clearer comparison, we multiply the PickScore, HPSv2.1, and CLIP scores by a factor of 100, ensuring readability while maintaining precision with four significant digits.

**Pick-a-Pic.**   The Pick-a-Pic dataset comprises a curated set of text-to-image pairs, meticulously gathered from user interactions within the Pick-a-Pic web application. Each entry in the dataset features a duo of images, accompanied by

a descriptive text prompt and a label that captures user preferences. This dataset is enriched with images produced by a spectrum of text-to-image generation models, such as Stable Diffusion 2.1, Dreamlike Photoreal 2.05, and several iterations of Stable Diffusion XL, each tested across a variety of CFG scales. For the purposes of this research, we focus on the training segment of the dataset. Moreover, to substantiate the robustness of our methodology, we include a detailed quantitative analysis of the test dataset in the supplementary materials, providing additional insights and validation of our findings.

**HPDv2.** The HPDv2 dataset is meticulously compiled by collecting a vast array of human preference data through the "Dreambot" channel on the Stable Foundation Discord server. This comprehensive dataset includes 25,205 unique text prompts and an impressive total of 98,807 images generated from these prompts. Each prompt is associated with several images and is annotated with preference labels that reflect user selections among image pairs. To rigorously evaluate the effectiveness of our approach, we employ a dedicated test dataset consisting of 3,200 text prompts. This test set plays a pivotal role in benchmarking the HPDv2 model's capability to accurately assess and predict human preferences in image evaluation tasks.

**Parti-Prompts.** Parti-Prompts is a meticulously curated collection of 1,632 text prompts specifically crafted to assess the efficacy of text-to-image generative models. This dataset encompasses a wide range of categories and integrates numerous intricate tasks aimed at comprehensively evaluating the models' generative prowess across various dimensions. The prompts are structured to provide a rigorous testing environment, facilitating an in-depth analysis of model performance. They are particularly effective in gauging the models' flexibility and precision when confronted with complex and diverse scenarios, thereby offering a thorough measure of their capabilities in real-world applications.

## E   User Study

Here, We perform a user study consisting of three questions: (1) Overall, which image do you consider to have the best quality? (2) Which image is the most visually appealing? (3) Which image has the highest alignment between text and image content?

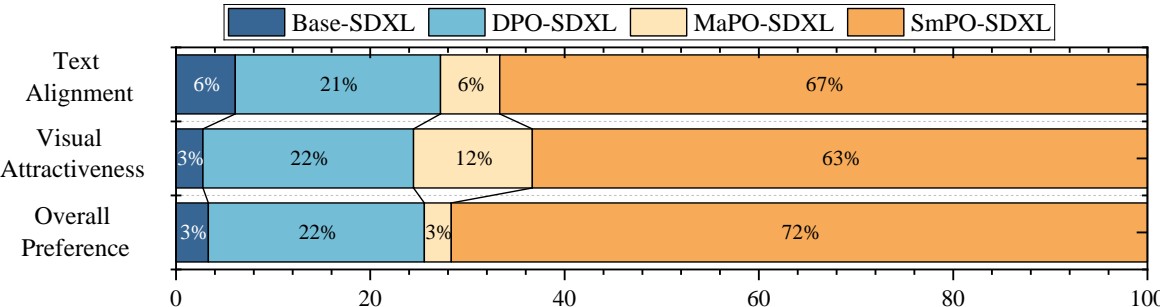

Figure 5: **User Studies.** We present the comparative user evaluation results of SmPO-SDXL against Base-SDXL, DPO-SDXL, and MaPO-SDXL. The first row displays the user study results for text alignment, the second row for visual attractiveness, and the third row for overall user preference. The results demonstrate that our method outperforms all existing baselines.

We randomly selected 50 prompts from Parti-Prompt and HPDv2 for image generation and invited 6 participants to evaluate the generated images based on three criteria. As illustrated in Figure 3, SmPO-SDXL achieved a 72% preference rate in overall evaluation, while DPO-SDXL, MaPO-SDXL, and Base-SDXL obtained 22%, 3%, and 3%, respectively. Furthermore, SmPO-SDXL demonstrated comparable performance in both visual Attractiveness and text alignment, with preference rates of 63% and 67%, respectively. Our results indicate that the SmPO fine-tuned SDXL base model significantly outperforms the baseline Base-SDXL model. Additionally, it exhibits superior performance compared to the advanced baselines, Diffusion-DPO and MaPO, across all evaluated metrics.

# F  Additional Quantitative Results

In this section, we provide additional quantitative evaluations.

- Table 8 presents the quantitative comparison against SDXL baselines on the Pick-a-Pic v2 test set.

- Table 8 presents the quantitative comparison against SD1.5 baselines on the Pick-a-Pic v2 test set.

The experimental outcomes indicate that our method outperforms the baseline models across nearly all reward-based evaluation metrics, highlighting its superior performance.

Table 8: **Additional quantitative comparison with SDXL baselines.** We apply the prompts from the Pick-a-Pic v2 test set to compare our model with the existing alignment baseline on the SDXL model. We report the median and mean values of five reward evaluators on the Pick-a-Pic v2 test set, retaining five significant figures. In the table, the highest value in each column is highlighted in **bold**, and the second highest is underlined. As shown, our model achieves the best results in nearly all reward evaluations.

| Baselines | PickScore | | HPSv2.1 | | ImageReward | | Aesthetic | | CLIP | |
|---|---|---|---|---|---|---|---|---|---|---|
| | Median | Mean | Median | Mean | Median | Mean | Median | Mean | Median | Mean |
| Base-SDXL | 22.226 | 22.153 | 28.482 | 28.102 | 0.7377 | 0.5622 | 5.9719 | 6.0061 | 36.512 | 36.138 |
| SFT-SDXL | 21.658 | 21.678 | 28.070 | 27.747 | 0.4851 | 0.3991 | 5.8693 | 5.8594 | 36.244 | 35.749 |
| DPO-SDXL | 22.600 | 22.627 | 29.708 | 29.612 | 1.0345 | 0.7993 | 6.0252 | 6.0168 | 37.386 | 37.376 |
| MaPO-SDXL | 22.200 | 22.272 | 29.251 | 28.978 | 1.1718 | 0.9135 | **6.2382** | **6.1941** | 36.628 | 36.072 |
| SmPO-SDXL | **23.123** | **23.052** | **32.173** | **31.691** | **1.3288** | **1.0742** | 6.1357 | 6.1259 | **37.634** | **37.381** |

Table 9: **Additional quantitative comparison with SD1.5 baselines.** We apply the prompts from the Pick-a-Pic v2 test set to compare our model with the existing alignment baseline on the SD1.5 model. We report the median and mean values of five reward evaluators on the Pick-a-Pic v2 test set, retaining five significant figures. In the table, the highest value in each column is highlighted in **bold**, and the second highest is underlined. As shown, our model achieves the best results in nearly all reward evaluations.

| Baselines | PickScore | | HPSv2.1 | | ImageReward | | Aesthetic | | CLIP | |
|---|---|---|---|---|---|---|---|---|---|---|
| | Median | Mean | Median | Mean | Median | Mean | Median | Mean | Median | Mean |
| Base-SD1.5 | 20.633 | 20.662 | 24.642 | 24.237 | -0.0865 | -0.1505 | 5.3334 | 5.3268 | 33.046 | 32.637 |
| SFT-SD1.5 | 21.184 | 21.255 | 27.949 | 27.728 | 0.6389 | 0.4771 | 5.6314 | 5.6220 | 34.261 | 34.055 |
| DPO-SD1.5 | 21.033 | 21.052 | 25.760 | 25.384 | 0.1653 | 0.0707 | 5.5254 | 5.4706 | 33.272 | 33.276 |
| KTO-SD1.5 | 21.200 | 21.193 | 27.917 | 27.719 | 0.6320 | 0.5395 | 5.5996 | 5.5820 | 33.996 | 33.940 |
| SmPO-SD1.5 | **21.628** | **21.576** | **28.944** | **28.507** | **0.8965** | **0.6401** | **5.7449** | **5.6997** | **35.020** | **34.864** |

