# OpenReview forum: "Smoothed Preference Optimization via ReNoise Inversion for Aligning Diffusion Models with Varied Human Preferences"
_ICML.cc/2025/Conference — ICML 2025 poster_

### Official Review · Reviewer_5qKt · 2025-03-10

**Overall Recommendation:** 4

**Summary:**

This paper proposes SmPO-Diffusion, a novel method for aligning text-to-image diffusion models with varied human preferences. The authors introduce two core contributions: (1) a smoothed preference modeling approach, replacing the binary preference distribution with a smooth distribution derived by reward models. (2) an optimization strategy called ReNoise Inversion, designed to estimate the dpo preference distribution by inversion and renoise.  Experimentally, the method achieves state-of-the-art results, surpassing existing baselines across multiple human preference evaluation metrics, and notably reduces training resource consumption.

**Claims And Evidence:**

Yes. The claims in the submission are supported by convincing evidence. The paper provides extensive experiments and ablation studies to support the claims.

**Essential References Not Discussed:**

No.

**Experimental Designs Or Analyses:**

Yes.
1. Primarily used the Pick-a-Pic v2 dataset and benchmarked against established baselines (e.g., Diffusion-DPO, MaPO, KTO, SFT), which is appropriate and valid.
2. Evaluated results using widely-accepted automatic metrics (e.g., PickScore, HPSv2.1), effectively capturing text-to-image quality.
3. Conducted detailed ablation studies clearly demonstrating the contributions of each method component; analyses were very convincing.
4. Included practical and valid computational efficiency analyses.

**Methods And Evaluation Criteria:**

Yes.

**Other Comments Or Suggestions:**

Please provide a clearer introduction and context for the compared methods, such as MaPO and Diffusion-KTO. Currently, these comparisons appear abruptly without prior explanation, making it confusing for readers. Clarifying why these particular methods were selected for comparison and briefly outlining their relevance to your work would significantly enhance readability and coherence.

**Other Strengths And Weaknesses:**

Strengths:

1. The proposed smoothed preference distribution appears novel and well-motivated. Incorporating DDIM inversion into DPO training is innovative, providing deeper insights into preference modeling and demonstrating notable originality.

2. The experimental results are convincing and impressive, clearly highlighting the significant potential of the proposed method.

3. The paper is clearly written and well-structured. The presented method achieves a good balance between simplicity and effectiveness.

Weaknesses:

1. While using automated reward models is practically advantageous, including explicit human validation results in the main text would considerably strengthen the claim that the proposed method enhances real-world alignment. Relying solely on reward models to capture fine-grained preference distributions appears overly simplistic, raising questions about why this method can outperform large-scale human preference datasets.

2. A key contribution of this paper is adapting DDIM inversion for sampling and noise injection within DPO training. However, additional theoretical justification is required. Specifically, it remains unclear why DDIM inversion is more effective than simple noise addition during training. Moreover, discussing whether this approach generalizes beyond DPO to other training paradigms would enhance the paper's impact.

**Questions For Authors:**

1. The idea of using DDIM inversion rather than simply adding noise during training is intriguing. Could this technique be generalized and applied effectively to other diffusion model training paradigms beyond DPO?

**Relation To Broader Scientific Literature:**

Yes.

1. The paper builds upon Direct Preference Optimization (DPO) techniques, extending recent methods such as Diffusion-DPO and MaPO, and addresses known issues like excessive optimization and misalignment.
2. The work utilizes DDIM and ReNoise inversion techniques from prior literature but uniquely applies them to preference optimization, demonstrating improved alignment and computational efficiency compared to earlier methods.

**Theoretical Claims:**

Yes.
I've checked the correctness of any proofs for theoretical claims.
The author provided the derivation of eq.9 in the appendix and it is correct.

---

> ### Author Rebuttal · Authors · 2025-03-29
>
> Thank you for highly recognizing the value of our study and helpful feedback!
>
> ---
>
> **Q1:** *Why do reward models outperform large human preference datasets despite their simplicity?*
>
> **A1:** This is a great question! We believe reward models have the following advantages:
>
> 1、We note that PickScore (reward model) feedback is interpretable as **pseudo-labeling** the Pick-a-Pic dataset—a form of data cleaning [1] [2]. Specifically, the PickScore reward model serves not only as a proxy for human preferences but also as a mechanism for **data refinement**.
>
> 2、**Granular Feedback vs. Binary Human Labels**
>
> Human preference datasets inherently provide *binary* labels (e.g., "A > B"), limiting their ability to represent nuanced preference distributions. In contrast, reward models assign **continuous scores**, enabling two key advantages:
>
> + **Smooth Supervision**: Loss functions can adaptively weight samples based on reward differences,avoiding the brittleness of binary thresholds.
>
> + **Relative Calibration**: Fine-grained rewards better capture perceptual similarity (e.g., distinguishing "slightly better" from "significantly better" cases), which is critical for guiding iterative model updates.
>
> This explains why reward-guided training can outperform raw human datasets: it amplifies signal-to-noise ratios in human preferences while retaining their semantic intent. We emphasize that our approach complements (not replaces) human feedback, as reward models are trained on human-annotated data.
>
> [1] Qizhe Xie, Minh-Thang Luong, Eduard Hovy, and Quoc V Le. Self-training with noisy student improves imagenet. CVPR 2020
>
> [2] Barret Zoph, Golnaz Ghiasi, Tsung-Yi Lin, Yin Cui, Hanxiao Liu, Ekin Dogus Cubuk, and Quoc Le. Rethinking pretraining and self-training. NeurIPS 2020
>
> ---
>
> **Q2:** *The paper uses DDIM inversion in DPO training but lacks theoretical justification of its advantage over basic noise addition.*
>
> **A2:**  Thank you for the valuable feedback! We first derive the minimal proof. Note that for Equ (5), Diffusion-DPO utilizes $q(x_{1:T}|x_{0})$ to approximate $p_{\theta}(x_{1:T}|x_{0})$. For each step, Diffusion-DPO used $q(x_{t-1,t}|x_{0})$ to approximate $p_{\theta}(x_{t-1,t}|x_{0})$. Suppose we replace standard noise injection at $x_{t-1}$ with **a single-step DDIM inversion**, when given $x_{0}$. Formally, we propose to use $q(x_{t-1}|x_{0})p_{\theta}(x_{t}|x_{t-1})$ for approximation. And this approximation yields lower error because $$D_{KL}(q(x_{t-1}|x_{0})p_{\theta}(x_{t}|x_{t-1})||p_{\theta}(x_{t-1,t}|x_{0})) = D_{KL}(q(x_{t-1}|x_{0})||p_{\theta}(x_{t-1}|x_{0})) <D_{KL}(q(x_{t-1,t}|x_{0})||p_{\theta}(x_{t-1,t}|x_{0})).$$
>
> The complete derivation employing full inversion will be detailed in the paper.
>
> ---
>
> **Q3:** *Could DDIM inversion be generalized and applied as an standard alternative to noise injection in diffusion training?*
>
> **A3:** Yes, generalizable!  We greatly appreciate the opportunity to discuss this point!
>
> Currently, the training of large-scale diffusion models/flow matching primarily consists of two stages: **pre-training** and **post-training**. The pre-training stage focuses on establishing the trajectory from a noise distribution to a data distribution, while we argue that the post-training stage should specifically fine-tune variables that are **highly correlated with the target images**.
>
> This idea shares some similarities with the **Reflow** method in **Rectified Flow** [3], which trains specialized (noise, image) pairs. However, a key distinction is that most real-world datasets contain only images (without paired noise). Thus, our motivation is to **adaptively adjust image-dependent variables** during post-training. We believe this approach could become a dominant paradigm in future post-training strategies, as it **preserves the model’s inherited capabilities while enhancing its task-specific performance**.
>
> [3] Liu, Xingchao, Chengyue Gong, and Qiang Liu. Flow straight and fast: Learning to generate and transfer data with rectified flow. ICLR2023
>
> ---
>
> **Q4:** *Please provide a clearer introduction and context for the compared baselines.*
>
> **A4:** Thank you for the suggestion! We fully agree that providing clearer motivation for method selection is critical for readability. Below, we outline our planned revisions to address this concern:
>
> These baselines share a common focus on **human-aligned image generation** but **differ fundamentally in their technical mechanisms**:  Diffusion-KTO adopts the Kahneman-Tversky [4] model to represent human utility instead of maximizing the log-likelihood of preferences. Margin-aware Preference Optimization (MaPO)  jointly maximizes the likelihood margin between the preferred and dispreferred datasets and the likelihood of the preferred sets, simultaneously learning general stylistic features and preference without reference model.
>
> [4] Tversky, A. and Kahneman, D. Advances in prospect theory: Cumulative representation of uncertainty.

---

### Official Review · Reviewer_HnYu · 2025-03-11

**Overall Recommendation:** 4

**Summary:**

This paper proposes a smoothed extension to DPO, where the preference data is smoothed to incorporate non-binary preference labels. The authors first created smoothed preference labels for image pairs using the likelihood estimation of a reward model. Then it uses noise-inversion to provide a better posterior estimation of the forward process during the optimization step. The author conducted experiments on a variety of models (SDXL, SD1.5) and showed that the proposed method outperforms existing baselines.

**Claims And Evidence:**

The author made two central claim: a) the existing framework based on binary feedbacks do not adequately address noise and inconsistencies in the human preference data, necessitates the incorporation of a smoothed relaxation. b)  the proposed noise inversion technique addresses the improves training process of preference alignment algorithm by providing better estimates of forward sampling trajectory. The author was able to support both of these claims with theoretical analysis and emprical experiments.

**Essential References Not Discussed:**

N/A

**Experimental Designs Or Analyses:**

See Methods And Evaluation Criteria section. The author employs a set of common setup for their experiments.

**Methods And Evaluation Criteria:**

The author employed common evaluation metrics (e.g. PickScore, HPSv2) and datasets (e.g. Parti-Prompts) that are widely used by related literature. The author also included relevant baselines such as Diffusion-DPO, Diffusion-KTO. The authors additionally conducted experiments on multiple models (SD1.5, and SDXL) to showcase the generalizability of the proposed method.  These results are comprehensive.

**Other Comments Or Suggestions:**

Table 7 caption can be improved, as it is not immediately clear if rows are reward model and columns are evaluation metric, or vice versa. It was only clearly after noticing the uparrow in column titles. I suggest the authors clearly state in the caption that rows represents different training signals while columns are evaluated metrics.

**Other Strengths And Weaknesses:**

1. The authors are suggested to discuss the statistic significance of the gap in table 7, 8 and table 9. These numerical metrics can be hard to interpret for people with limited exposure to prior literature, as they have drastically different scales and some gap may be perceived as "insignificant".

**Questions For Authors:**

N/A

**Relation To Broader Scientific Literature:**

The proposed work proposed a novel solution to a problem that is well-recognized by the community (i.e. noises and inconsistencies of working with human preference data).  This work provide meaningful insight to the problem and can inspire future solutions.

**Theoretical Claims:**

I checked derivation in the main paper and appendix, they look good to me.

---

> ### Author Rebuttal · Authors · 2025-03-29
>
> We're truly grateful for your enthusiastic reception of our manuscript and your insightful feedback!
>
> ---
>
> **Q1：** *Table 7 caption can be improved.*
>
> **A1：** We sincerely appreciate your constructive feedback! We have implemented the following improvements:
>
> + **Caption Revision**: We have revised the table caption to explicitly state:
>   **"Table 7: Impact of weight-to-sensitivity ratio using different reward models. We report the performance comparison where rows represent different training signals and columns denote evaluation metrics. Upward arrows (↑) in column headers indicate higher values are preferable."**
> + **Visual Reinforcement**: To further enhance clarity: We will include a brief structural description in the Results section (Section 5.4) when first referencing the table in "Choice of Reward Model" subsection.
>
> ---
>
> **Q2**：*Discuss the statistic significance of the gap in table 7, 8, 9.*
>
> **A2:**  Thank you for this helpful suggestion!
>
> To provide a comprehensive statistical evaluation of our approach, we present both **win-rate analysis** and **effect size measurements**.
>
> + The win-rate results in *Supp Table 1-2* demonstrate how frequently evaluators prefer SmPO generations over baseline methods, with values exceeding **50**% (indicating statistical majority preference) highlighted in bold.
>
> Supp Table 1: Win-rate comparison between SmPO-SDXL and baselines on Pick-a-Pic v2 test set.
>
> |               | PickScore | HPSv2.1  | ImageReward | Aesthetic | CLIP     |
> | ------------- | --------- | -------- | ----------- | --------- | -------- |
> | vs. SDXL      | **89.6**  | **93.6** | **81.5**    | **65.5**  | **60.6** |
> | vs. SFT-SDXL  | **96.6**  | **94.2** | **81.3**    | **76.3**  | **67.3** |
> | vs. DPO-SDXL  | **75.7**  | **85.9** | **71.9**    | **63.9**  | **55.4** |
> | vs. MaPO-SDXL | **83.5**  | **86.5** | **61.0**    | 47.4      | **60.2** |
>
>
>
> Supp Table 2: Win-rate comparison between SmPO-SD1.5 and baselines on Pick-a-Pic v2 test set.
>
> |               | PickScore | HPSv2.1  | ImageReward | Aesthetic | CLIP     |
> | ------------- | --------- | -------- | ----------- | --------- | -------- |
> | vs. SD1.5     | **82.5**  | **88.0** | **80.1**    | **79.3**  | **66.5** |
> | vs. SFT-SD1.5 | **68.5**  | **66.1** | **62.2**    | **59.0**  | **60.0** |
> | vs. DPO-SD1.5 | **72.3**  | **83.7** | **76.1**    | **69.7**  | **63.3** |
> | vs. KTO-SD1.5 | **68.3**  | **67.3** | **57.0**    | **64.7**  | **59.8** |
>
>
>
> + Additionally, we compute **Cohen's d** to quantify the **effect sizes** between SmPO and baselines in *Supp Table 3-6*, following conventional interpretations: |d|<0.2 (small), 0.2≤|d|<0.5 (medium), 0.5≤|d|<0.8 (large), and |d|≥0.8 (very large). **Cohen's d measures the standardized mean difference in standard deviation units, making it unit-free.**
>
>
>
> Supp Table 3: Cohen's d for comparison between SmPO-SDXL and baselines on Pick-a-Pic v2 test set.
>
> |               | PickScore | HPSv2.1 | ImageReward | Aesthetic | CLIP  |
> | ------------- | --------- | ------- | ----------- | --------- | ----- |
> | vs. SDXL      | 0.600     | 0.860   | 0.537       | 0.205     | 0.203 |
> | vs. SFT-SDXL  | 0.937     | 0.960   | 0.693       | 0.480     | 0.266 |
> | vs. DPO-SDXL  | 0.282     | 0.505   | 0.290       | 0.192     | 0.010 |
> | vs. MaPO-SDXL | 0.508     | 0.648   | 0.176       | -0.120    | 0.213 |
>
>
>
> Supp Table 4: Cohen's d for comparison between SmPO-SD1.5 and baselines on Pick-a-Pic v2 test set.
>
> |               | PickScore | HPSv2.1 | ImageReward | Aesthetic | CLIP  |
> | ------------- | --------- | ------- | ----------- | --------- | ----- |
> | vs. SD1.5     | 0.650     | 1.010   | 0.730       | 0.652     | 0.385 |
> | vs. SFT-SD1.5 | 0.227     | 0.188   | 0.157       | 0.136     | 0.139 |
> | vs. DPO-SD1.5 | 0.368     | 0.732   | 0.529       | 0.392     | 0.275 |
> | vs. KTO-SD1.5 | 0.272     | 0.192   | 0.099       | 0.209     | 0.161 |
>
>
>
> Supp Table 5: Cohen's d for comparison between SmPO-SDXL and baselines on HPD v2 test set.
>
> |               | PickScore | HPSv2.1 | ImageReward | Aesthetic | CLIP  |
> | ------------- | --------- | ------- | ----------- | --------- | ----- |
> | vs. SDXL      | 0.601     | 0.914   | 0.498       | 0.206     | 0.124 |
> | vs. SFT-SDXL  | 1.084     | 0.957   | 0.647       | 0.419     | 0.272 |
> | vs. DPO-SDXL  | 0.336     | 0.527   | 0.192       | 0.219     | 0.012 |
> | vs. MaPO-SDXL | 0.537     | 0.772   | 0.162       | 0.021     | 0.149 |
>
>
>
> Supp Table 6: Cohen's d for comparison between SmPO-SD1.5 and baselines on HPD v2 test set.
>
> |               | PickScore | HPSv2.1 | ImageReward | Aesthetic | CLIP  |
> | ------------- | --------- | ------- | ----------- | --------- | ----- |
> | vs. SD1.5     | 1.007     | 1.360   | 0.946       | 0.778     | 0.473 |
> | vs. SFT-SD1.5 | 0.362     | 0.181   | 0.165       | 0.201     | 0.185 |
> | vs. DPO-SD1.5 | 0.641     | 0.979   | 0.686       | 0.481     | 0.327 |
> | vs. KTO-SD1.5 | 0.413     | 0.246   | 0.173       | 0.244     | 0.246 |

---

> > ### Comment · Reviewer_HnYu · 2025-04-02
> >
> > Thanks for the response. I keep my recommendation for acceptance

---

### Official Review · Reviewer_4khg · 2025-03-14

**Overall Recommendation:** 1

**Summary:**

The paper introduces SmPO-Diffusion, an approach for aligning text-to-image diffusion models with AI preferences by refining the Direct Preference Optimization framework. Instead of using a binary preference, the authors propose a smoothed preference distribution based on a reward model.

**Claims And Evidence:**

1. Smoothed Preference Modeling. Smoothed Labeling has been used in DPO in 2023. See *Essential References Not Discussed*.

2. Optimization via Renoise Inversion. There is no optimality analysis.

**Essential References Not Discussed:**

1. Lack of comparison and discussion with many Diffusion DPO variants.

2. Label smoothing has been proposed by the DPO authors in 2023 [1].

[1] A note on DPO with noisy preferences & relationship to IPO. https://ericmitchell.ai/cdpo.pdf

**Experimental Designs Or Analyses:**

The ablation study on hyperparameters are not comprehensive.

**Methods And Evaluation Criteria:**

For evaluation, Image Reward, and Aesthetic score could also be considered as benchmarks.

**Other Comments Or Suggestions:**

N/A

**Other Strengths And Weaknesses:**

N/A

**Questions For Authors:**

1. I cannot understand what the contribution is but adding more hyperparameters in the loss function. I would more than appreciate it if the authors can clarify it.

2. It seems that $\gamma$ can be absorb in $\beta$ by $({2\alpha} - \gamma)\beta = (\frac{2\alpha}{\gamma} - 1)\gamma\beta$ and $\frac{\alpha}{\gamma}$ is calculated by equation (12).

**Relation To Broader Scientific Literature:**

I cannot understand what the contribution is in this paper. It seems that this paper changes the hyperparameter $\beta$ to $(2\alpha -\gamma)\beta$, and tunes the hypermeters on specific datasets. That is all.

**Theoretical Claims:**

There is no formal proof in this paper.

---

> ### Author Rebuttal · Authors · 2025-03-29
>
> Thank you for your feedback and we'll do our utmost to resolve your concerns.
>
> ---
>
> **Q1:**  *For evaluation, Image Reward, and Aesthetic score could also be considered.*
>
> **A1:**  We have incorporated both metrics: **Image Reward** and **Aesthetic Score**  are reported in **Table 2,8 and 9** (Quantitive comparison), **Table 3-7** (Ablation studies) and discussed in Line 262-263 of **Section 5.1**.
>
> ---
>
> **Q2:** *There is no formal proof in this paper.*
>
> **A2:** The detailed derivation of SmPO loss is provided in **Appendix B**.
>
> ---
>
> **Q3:** *The ablation study on hyperparameters are not comprehensive.*
>
> **A3:** Our **ablation studies** on hyperparameters were systematically designed along five key dimensions to validate our method’s robustness:
>
> + **Component Effectiveness: Table 3** shows each module's contribution through progressive ablation.
> + **DDIM Inversion Steps: Table 4** shows step selection's impact on output quality and training efficiency.
> + **$(\gamma,\beta)$ combinations : Table 5** shows hyperparameter interaction effects.
> + **CFG in Inversion: Table 6** shows CFG's influence during DDIM inversion.
> + **Weight-to-Sensitivity Ratio $\frac{\alpha}{\gamma}$: Table 7** shows reward-model derived calculations.
>
> ---
>
> **Q4:** *Lack of comparison and discussion with many Diffusion DPO variants.*
>
> **A4:** Thank you for your feedback. We've compared our method with key baselines: **SFT, Diffusion-DPO,** its variants (**Diffusion-KTO and MaPO**). Additional **SPO** comparison are included (see response **A3** to Reviewer UNNR). To our knowledge, these are the most relevant and publicly available baselines. We welcome suggestions for additional baselines with references.
>
> ---
>
> **Q5:** *Label smoothing has been proposed by the DPO authors in 2023, Conservative DPO (cDPO).*
>
> **A5:** Thank you for your feedback. In fact, cDPO **differs** fundamentally from our proposed SmPO.
>
> 1. cDPO focuses on noisy labels (where labels may be **flipped** with some probability) and applies a **linear weighting** of swapped DPO losses: $L_{cDPO}(x_0^{w},x_0^{l})=(1-\epsilon)L_{DPO}(x_0^{w},x_0^{l})+\epsilon L_{DPO}(x_0^{l},x_0^{w})$. Our SmPO assigns **fine-grained** labels to each image pair and incorporates them into the DPO loss function through distribution averaging, yielding Equ (9).
> 2. In cDPO, $\epsilon$ is a is a **fixed** manually-tuned hyperparameter (refer to Line 50 and 84 of [1]), whereas our SmPO introduces reward-adaptive smoothing - zeroing loss for similar pairs, amplifying gradients for dissimilar ones. In other words, distinct image pairs are assigned different weights. As such, in Equ (12), $\frac{\alpha}{\gamma}$ could be more precisely represented as $\frac{\alpha}{\gamma}(x_{0}^{w},x_{0}^{l})$.
>
> [1] https://github.com/eric-mitchell/direct-preference-optimization/blob/main/trainers.py
>
> ---
>
> **Q6:** *I cannot understand what the contribution is.*
>
> **A6:** We appreciate your feedback and are happy to clarify our key contributions:
>
> 1. **Smoothed Preference:** We propose a novel *smoothed preference distribution* replacing DPO's binary modeling. Unlike fixed-weight approaches, our *(2α-γ)* scaling factor acts as a **pair-dependent dynamic regulator**, automatically driving loss to zero for similar pairs while amplifying gradients for dissimilar ones with each image pair's reward signal.
>
> 2. **Precision Optimization:**
>    Unlike Diffusion-DPO's random noise injection, our *Renoise Inversion* technique enables **direct optimization of image-correlated variables.** This contribution provides more stable training and higher efficiency.
>
> ---
>
> **Q7:** *It seems that $\gamma$ can be absorb in $\beta$ by $(2\alpha-\gamma)\beta=(\frac{2\alpha}{\gamma}-1)\gamma\beta$ is calculated by equation (12).*
>
> **A7:** We appreciate this observation.
>
> The term $(2\alpha-\gamma)$ is *image-pair-dependent*, automatically adjusted based on reward differences.  The core idea of $(2\alpha-\gamma)$ is to ensure that *"the loss decreases when preferences are more similar, and increases otherwise."*  **This can be implemented via different parameterization strategies.** For example, $\gamma$ can be step- or epoch-aware, adjusting dynamically to avoid being absorbed into $\beta$. We adopt a simple formulation: $(2\alpha-\gamma)$ is implemented as $(\frac{2\alpha}{\gamma}-1)\gamma$, balancing adaptability and numerical stability.  Ablations (Table 5) validate the impact of different $(\gamma,\beta)$ pair configurations.
>
> ---
>
> **Q8:** *Optimization via Renoise Inversion. There is no optimality analysis.*
>
> **A8:**  Thank you for your feedback. Optimizing Equ  (10) requires sampling $x_{1:T}\sim p_{\theta}^{c}(x_{1:T}|x_{0})$. While Diffusion-DPO approximates this with $q(x_{1:T}|x_{0})$, we argue precise $x_{0}$ reconstruction needs more accurate latents. This motivates our use of diffusion reconstruction methods (DDIM Inversion), supported by additional theoretical analysis (refer to response **A2** to Reviewer 5qKt).

---

### Official Review · Reviewer_UNNR · 2025-03-19

**Overall Recommendation:** 3

**Summary:**

This paper proposes a post-training method for diffusion models, named SmPO, which is modified from diffusion-dpo. SmPO recognize the variability of human preferences by replacing binary preferences with smoothed preference distributions, thereby mitigating label bias. In addition, Renoise Inversion method is employed to estimate the sampling trajectory. Compared to previous methods that randomly sample noise from a Gaussian distribution, this inversion method provides a more accurate estimation of the trajectory preference distribution. Extensive experiments demonstrate the strong performance of SmPO and the effectiveness of the proposed modules.

**Claims And Evidence:**

Claim: Human preference is variable. Simply considering binary preference of a pair of image causes label bias.

Evidence: Exp in Table 3




Claim: inversion based method can provide better estimation of the trajectory preference distribution.

Evidence: Exp in Table 3

**Essential References Not Discussed:**

I believe the following papers should be mentioned in the related works section or even compared against.

[1] Deng, Fei, et al. "Prdp: Proximal reward difference prediction for large-scale reward finetuning of diffusion models." Proceedings of the IEEE/CVF Conference on Computer Vision and Pattern Recognition. 2024.

[2] Liang, Zhanhao, et al. "Step-aware preference optimization: Aligning preference with denoising performance at each step." arXiv preprint arXiv:2406.04314 2.5 (2024): 7.

[3] Karthik, Shyamgopal, et al. "Scalable ranked preference optimization for text-to-image generation." arXiv preprint arXiv:2410.18013 (2024).

**Experimental Designs Or Analyses:**

1. Computational cost comparison in Table 1: The GPU hours for SmPO-SDXL should include the training time of the reward model, as other methods rely only on preference pair data, whereas SmPO-SDXL requires a reward model.

2. There are no experiments validating the design of the weight-to-sensitivity ratio.

3. In the qualitative results, the image-text alignment appears to be improved. Could you provide quantitative results on the GenEval [1] benchmark to verify this?

[1] Ghosh, Dhruba, Hannaneh Hajishirzi, and Ludwig Schmidt. "Geneval: An object-focused framework for evaluating text-to-image alignment." Advances in Neural Information Processing Systems 36 (2023): 52132-52152.

**Methods And Evaluation Criteria:**

yes

**Other Comments Or Suggestions:**

I will modify the score according to the opinions of other reviewers and the author's response.

**Other Strengths And Weaknesses:**

The rationale for designing the weight-to-sensitivity ratio as given in Equation 12 is unclear.

**Questions For Authors:**

None

**Relation To Broader Scientific Literature:**

1. The standard practice of previous dpo-based methods [1,2,3] is to use binary label. This paper proposes a new approach that instead employs smoothed preference distributions and demonstrates the effectiveness of this replacement through experiments

2. While using q(x_{1:T}|x_0) to approximate the reverse process is proposed by [1] and is widely accepted by the community, this paper proposes using inversion to achieve a more accurate estimation.

[1] Wallace, Bram, et al. "Diffusion model alignment using direct preference optimization." Proceedings of the IEEE/CVF Conference on Computer Vision and Pattern Recognition. 2024.

[2] Yang, Kai, et al. "Using human feedback to fine-tune diffusion models without any reward model." Proceedings of the IEEE/CVF Conference on Computer Vision and Pattern Recognition. 2024.

[3] Yang, Shentao, Tianqi Chen, and Mingyuan Zhou. "A dense reward view on aligning text-to-image diffusion with preference." arXiv preprint arXiv:2402.08265 (2024).

**Theoretical Claims:**

I have reviewed the derivation of the loss function in equation 16, and it appears correct to me. However, I will also consider the opinions of other reviewers.

---

> ### Author Rebuttal · Authors · 2025-03-29
>
> We are honored by your favorable evaluation and have carefully considered your suggestions!
>
> ---
>
> **Q1**: *The rationale for designing the weight-to-sensitivity ratio as given in Equ (12) is unclear.*
>
> **A1**: Thank you for your feedback!  According to Equ (8), $\tilde{p}(x_{0}^{w}|c) = \frac{p(x_{0}^{w}|c)^{\alpha}p(x_{0}^{l}|c)^{\gamma-\alpha}}{Z_{p}^{w}(c)}=\frac{(p(x_{0}^{w}|c)^{\frac{\alpha}{\gamma}}p(x_{0}^{l}|c)^{1-\frac{\alpha}{\gamma}})^{\gamma}}{Z_{p}^{w}(c)}$
>
> where weight-to-sensitivity ratio could be regarded as the pairwise preference probability $p(x_{0}^{w} \succ x_{0}^{l}|c)$. Since a pairwise preference is hard to model directly, we adopt the well-established Bradley-Terry framework through the reward model as Equ (12).
>
> ---
>
> **Q2:** *The GPU hours for SmPO should include the training time of the reward model.*
>
> **A2**: Thank you for the feedback! Training PickScore requires **8×A800 GPUs for ~40 minutes** [1]. We have updated **Table 1** to include this cost.
>
> Supp Table 1: Computational cost comparison
>
> | Model     | GPU Hours              |
> | --------- | ---------------------- |
> | DPO-SDXL  | ~976.0                 |
> | MaPO-SDXL | ~834.4                 |
> | SmPO-SDXL | ~**150.8 (145.5+5.3)** |
>
> | Model      | GPU Hours            |
> | ---------- | -------------------- |
> | DPO-SD1.5  | ~204.8               |
> | KTO-SD1.5  | ~1056.0              |
> | SmPO-SD1.5 | ~**41.3 (36.0+5.3)** |
>
> [1] https://github.com/yuvalkirstain/PickScore
>
> ---
>
> **Q3:** *PRDP, SPO and RankDPO should be included.*
>
> **A3:** We appreciate the suggestion! We'll update our related work with these papers and include the SmPO-SPO comparison.
>
> Supp Table 2: Median score comparison of SD1.5 on HPD v2.
>
> |            | PickScore | HPSv2.1   | ImageReward | Aesthetic | CLIP      |
> | ---------- | --------- | --------- | ----------- | --------- | --------- |
> | SPO-SD1.5  | 21.49     | 26.74     | 0.181       | 5.655     | 32.513    |
> | SmPO-SD1.5 | **23.62** | **32.53** | **1.331**   | **6.264** | **38.88** |
>
> Supp Table 3: Win-rate comparison on Parti-Prompts.
>
> |                         | PickScore | HPSv2.1 | ImReward | Aesthetic | CLIP  |
> | ----------------------- | --------- | ------- | -------- | --------- | ----- |
> | SmPO-SD1.5 vs SPO-SD1.5 | 70.81     | 75.47   | 76.62    | 65.03     | 79.96 |
> | SmPO-SDXL vs SPO-SDXL   | 58.37     | 52.44   | 63.72    | 52.36     | 74.25 |
>
> **However, Comparisons with PRDP (different SD-v1.4 base) and RankDPO (different training data) are limited by the unavailability of checkpoints.**
>
> ---
>
> **Q4:** *There are no experiments validating the design of the weight-to-sensitivity ratio.*
>
> **A4:**  Thank you for the feedback. We have designed two experiments to verify this aspect:
>
> 1. **Ablation Study (Table 3):**
>    The results demonstrate that incorporating our PickScore-based weight-to-sensitivity ratio (*+Smoothed*) leads to **significant performance gains**, validating its effectiveness.
> 2. **Reward Model Analysis (Table 7):**
>    We have further compared the weight-to-sensitivity ratio with different reward models, providing additional empirical support for our design choice.
>
> ---
>
> **Q5:**  *Could you provide quantitative results on the GenEval benchmark?*
>
> **A5:**  Yes! Thank you for highlighting this important aspect. We have conducted quantitative results on the GenEval.
>
> Supp Table 4: GenEval scores over SD1.5 baselines.
>
> | Model | single object | two object | counting | Attribute binding | position | colors   | overall  |
> | ----- | ------------- | ---------- | -------- | ----------------- | -------- | -------- | -------- |
> | SD1.5 | **0.96**      | 0.38       | 0.35     | 0.04              | 0.03     | 0.76     | 0.42     |
> | DPO   | **0.96**      | 0.40       | 0.38     | 0.05              | 0.04     | 0.77     | 0.43     |
> | KTO   | **0.96**      | 0.44       | 0.39     | **0.08**          | 0.07     | 0.78     | 0.45     |
> | SPO   | **0.96**      | 0.35       | 0.36     | 0.06              | 0.05     | 0.77     | 0.43     |
> | SmPO  | **0.96**      | **0.52**   | **0.40** | **0.08**          | **0.08** | **0.80** | **0.47** |
>
>
> Supp Table 5: GenEval scores over SDXL baselines.
>
> | Model | single object | two object | counting | Attribute binding | position | colors   | overall  |
> | ----- | ------------- | ---------- | -------- | ----------------- | -------- | -------- | -------- |
> | SDXL  | 0.97          | 0.70       | 0.41     | 0.22              | 0.10     | 0.87     | 0.55     |
> | DPO   | 0.98          | **0.80**   | 0.45     | 0.24              | 0.11     | 0.88     | 0.58     |
> | MaPO  | 0.95          | 0.70       | 0.36     | 0.26              | 0.10     | 0.88     | 0.54     |
> | SPO   | 0.97          | 0.73       | 0.38     | 0.17              | 0.10     | 0.86     | 0.53     |
> | SmPO  | **0.99**      | **0.80**   | **0.46** | **0.30**          | **0.15** | **0.89** | **0.60** |
>
> Experimental results indicate that SmPO consistently enhances image-text alignment performance.

---

### Decision · Program_Chairs · 2025-05-01

**Decision:**

Accept (poster)

**Comment:**

This paper investigates preferences optimization for T2I diffusion models and introduces an alternative to the binary reward assignment in their method, SmPO-Diffusion. It is argued that the current preference labeling scheme lacks granularity and thus calls for a smooth reward modeling. In addition, Renoise Inversion is introduced to estimate the sampling trajectory.

The paper is rated by the reviewers as WA, R, A, A. While there are concerns over the lack of comparison to related work (e.g., PRDP, SPO and RankDPO) and insufficient study on hyperparameters, the overall feeling on the paper remains positive, evidenced by the recognition of the strong experimental results, the well motivated method design and the theoretical analysis for support, and the paper being written and well-structured. Reviewer 4khg raised a few concerns such as similarity to label smoothing, lack of proof, and lack comparisons with existing methods. The author rebuttal addressed these concerns.

Overall, the AC recommends accept and encourages the authors to fully address the mention issues by the reviewers in their final version.